# The Well: a Large-Scale Collection of Diverse Physics Simulations for Machine Learning

**Ruben Ohana** [1,2,*], **Michael McCabe** [1,*], **Lucas Meyer** [1], **Rudy Morel** [1,2],
**Fruzsina J. Agocs** [2,3,†], **Miguel Beneitez** [4,†], **Marsha Berger** [2,5,†], **Blakesley Burkhart** [2,6,†],
**Stuart B. Dalziel** [4,†], **Drummond B. Fielding** [2,7,†], **Daniel Fortunato** [2,†],
**Jared A. Goldberg** [2,†], **Keiya Hirashima** [1,2,8,†], **Yan-Fei Jiang** [2,†], **Rich R. Kerswell** [4,†],
**Suryanarayana Maddu** [2,†], **Jonah Miller**[9,†], **Payel Mukhopadhyay** [10,†], **Stefan S. Nixon** [4,†],
**Jeff Shen** [11,†], **Romain Watteaux** [12,†], **Bruno Régaldo-Saint Blancard** [1,2],
**François Rozet**[1,13], **Liam H. Parker**[1,2,10], **Miles Cranmer** [1,4], **Shirley Ho** [1,2,5,11]

[1] Polymathic AI [2] Flatiron Institute [3] University of Colorado, Boulder [4] University of Cambridge
[5] New York University [6] Rutgers University [7] Cornell University [8] University of Tokyo
[9] Los Alamos National Laboratory [10] University of California, Berkeley
[11] Princeton University [12] CEA DAM [13] University of Liège

## Abstract

Machine learning based surrogate models offer researchers powerful tools for accelerating simulation-based workflows. However, as standard datasets in this space often cover small classes of physical behavior, it can be difficult to evaluate the efficacy of new approaches. To address this gap, we introduce *the Well:* a large-scale collection of datasets containing numerical simulations of a wide variety of spatiotemporal physical systems. The Well draws from domain experts and numerical software developers to provide 15TB of data across 16 datasets covering diverse domains such as biological systems, fluid dynamics, acoustic scattering, as well as magneto-hydrodynamic simulations of extra-galactic fluids or supernova explosions. These datasets can be used individually or as part of a broader benchmark suite. To facilitate usage of the Well, we provide a unified PyTorch interface for training and evaluating models. We demonstrate the function of this library by introducing example baselines that highlight the new challenges posed by the complex dynamics of the Well. The code and data is available at `https://github.com/PolymathicAI/the_well`.

## 1 Introduction

Simulation is one of the most ubiquitous and important tools in the modern computational science and engineering toolbox. From forecasting [1–3], to optimization [4, 5], to parameter inference [6, 7], practitioners lean heavily on simulation to evaluate how physical systems will evolve over time in response to varying initial conditions or stimuli. For many physical phenomena, this evolution can be described by systems of *partial differential equations* (PDEs) which model fundamental physical behavior aggregated to the continuum level under different material assumptions. Unfortunately, finding analytical solutions is infeasible for all but restricted classes of PDEs [8]. As a result, *numerical methods* which solve discretized versions of these equations with well-understood convergence and approximation properties have become the preeminent approach in this space. However, in some cases, numerical methods can provide accuracy in excess of what is needed for applications at significant computational cost while lower resolution direct simulation may not resolve key features of the dynamics. This has spurred the development of faster, simplified models referred to as *surrogate models* that resolve only the essential features for a given scale of simulation [9, 10].

---

[*]Equal contribution. Contact: `{rohana, mmccabe}@flatironinstitute.org`
[†]Domain expert, alphabetical order.

38th Conference on Neural Information Processing Systems (NeurIPS 2024) Track on Datasets and Benchmarks.

It is this surrogate modeling space where deep learning is poised to make a significant impact [11–13] with tangible results already demonstrated across diverse sets of fields and applications [3, 14–18]. Yet despite these successes, deep learning based surrogates face significant challenges in reaching broader adoption. One reason for this is the gap between the complexity of problems of practical interest and the datasets used for evaluating these models today. Scaling analysis has shown that deep learning-based surrogates can require large amounts of data to reach high accuracy [19, 20]. Meanwhile, even at resolutions accessible to modern machine learning architectures, high-quality scientific simulation can require the combination of specialized software, domain expertise, and months of supercomputer time [21]. On the other hand, from the perspective of scientists running these simulations, even just storing the frequent snapshots necessary for conventional off-line deep learning training is a significant and unnecessary expense [22–24]).

To address this gap, we introduce *the Well*, a diverse 15 TB collection of high quality numerical simulations produced in close collaboration with domain experts and numerical software developers. The Well is curated to provide challenging learning tasks at a scale that is approachable to modern machine learning but where efficiency remains an important concern. It contains 16 datasets ranging across application domains, scales, and governing equations from the evolution of biological systems to the growth of galaxies. Each dataset contains temporally coarsened snapshots from simulations of a particular physical phenomenon across multiple initial conditions or physical parameters, while providing a sufficiently large number of snapshots to explore simulation stability. Furthermore, the Well provides machine learning researchers with complex, demanding benchmarks that will inform the development of the next generation of data-driven surrogates.

## 2   Related Work

Modern machine learning relies on massive, curated and diverse datasets [25–28]. Natural language processing is built on internet-scale datasets [29–32], while vision models have grown to utilize sets containing billions of text-images pairs [33]. These datasets are sufficiently diverse that model improvement can be derived from sophisticated filtering approaches [32, 34, 35].

On the other hand, datasets designed for physical dynamics prediction are still growing. Early datasets featured a variety of common reference simulations [36–38]. While these datasets have seen rapid adoption, the broader trend has moved towards more complex but specialized simulation datasets [39–45]. These have opened new application areas for deep learning but have typically been limited to a small number of tasks. Other datasets have tackled more ambitious high-resolution problems [46–48], but the limited number of snapshots and scale of individual samples often restricts their usage. New datasets which offer complexity, volume, and diversity simultaneously are necessary for holistic evaluation of individual models and for the emerging trend of multiple physics foundation models [49–54]. The Well provides unified access to a collection of physical scenarios and benchmarking tools that are both diverse and challenging.

## 3   Diving into the Well

While the Well can be used for many tasks, several of which are highlighted in Appendix D, the one we focus on is *surrogate modeling*. Surrogate models estimate a solution function $\hat{U}(x,t)$ to a partial differential equation from some basic inputs, most commonly initial conditions $U(x,0)$ and/or boundary conditions. This is often, but not always, cast as an autoregressive prediction problem where time is discretized into samples at $t \in \{t_1, t_2, ..., t_T\}$ and model $f$ is then trained to predict:

$$\hat{U}(x,t_{i+1}) = f(\hat{U}(x,t_i))$$

where $\hat{U}(x,0) = U(x,0)$ until the solution estimate has been generated for all $t$. We note that for 2D data with a uniform spatial discretization, this process closely resembles video generation.

**Format.** The Well is composed of 16 datasets totaling 15TB of data with individual datasets ranging from 6.9GB to 5.1TB. The data is provided on uniform grids and sampled at constant time intervals. Data and associated metadata are stored in self-documenting `HDF5` files [55]. All datasets use a shared data specification described in the supplementary materials and a PyTorch [56] interface is provided. These files include all available state variables or spatially varying coefficients associated with a given set of dynamics in `numpy` [57] arrays of shape `(n_traj, n_steps, coord1, coord2, (coord3))` in single precision `fp32`. We distinguish between scalar, vector, and tensor-valued fields due to their different transformation properties. Each file is randomly split into training/testing/validation sets with a respective split of 0.8/0.1/0.1 * `n_traj`. Details of individual datasets are given in Table 1.

**Extensibility.** The PyTorch interface can process any data file following the provided specification without any additional modification to the code base. Scripts are provided to check whether HDF5 files are formatted correctly. This allows users to easily incorporate third-party datasets into pipelines using the provided benchmarking library.

| Dataset | CS | Resolution (pixels) | n_steps | n_traj |
|---|---|---|---|---|
| acoustic_scattering | $(x,y)$ | $256 \times 256$ | 100 | 8000 |
| active_matter | $(x,y)$ | $256 \times 256$ | 81 | 360 |
| convective_envelope_rsg | $(r,\theta,\phi)$ | $256 \times 128 \times 256$ | 100 | 29 |
| euler_multi_quadrants | $(x,y)$ | $512 \times 512$ | 100 | 10000 |
| gray_scott_reaction_diffusion | $(x,y)$ | $128 \times 128$ | 1001 | 1200 |
| helmholtz_staircase | $(x,y)$ | $1024 \times 256$ | 50 | 512 |
| MHD | $(x,y,z)$ | $64^3$ and $256^3$ | 100 | 100 |
| planetswe | $(\theta,\phi)$ | $256 \times 512$ | 1008 | 120 |
| post_neutron_star_merger | $(\log r,\theta,\phi)$ | $192 \times 128 \times 66$ | 181 | 8 |
| rayleigh_benard | $(x,y)$ | $512 \times 128$ | 200 | 1750 |
| rayleigh_taylor_instability | $(x,y,z)$ | $128 \times 128 \times 128$ | 120 | 45 |
| shear_flow | $(x,y)$ | $256 \times 512$ | 200 | 1120 |
| supernova_explosion | $(x,y,z)$ | $64^3$ and $128^3$ | 59 | 1000 |
| turbulence_gravity_cooling | $(x,y,z)$ | $64 \times 64 \times 64$ | 50 | 2700 |
| turbulent_radiative_layer_2D | $(x,y)$ | $128 \times 384$ | 101 | 90 |
| turbulent_radiative_layer_3D | $(x,y,z)$ | $128 \times 128 \times 256$ | 101 | 90 |
| viscoelastic_instability | $(x,y)$ | $512 \times 512$ | variable | 260 |

Table 1: Dataset description: coordinate system (CS), resolution of snapshots, n_steps (number of time-steps per trajectory), n_traj (total number of trajectories in the dataset).

## 3.1 Contents of the Well

This section provides physical intuition and background for the scenarios contained in the datasets along with visualizations in Figures 1–5. Technical details on the underlying physics, fields, physical parameters, and the generating processes for the datasets are given in Appendix C.

### 3.1.1 acoustic_scattering

Acoustic scattering possesses simple linear dynamics that are complicated by the underlying geometry. In this dataset, we model the propagation of acoustic waves through a domain consisting of substrata with sharply variable density in the form of maze-like walls (Figure 1, top) or pockets with vastly differing compositions. These simulations are most commonly seen in inverse problems including source optimization and inverse acoustic scattering in which sound waves are used to probe the composition of the domain.

### 3.1.2 active_matter

Active matter systems are composed of agents, such as particles or macromolecules, that transform chemical energy into mechanical work, generating active forces or stresses. These forces are transmitted throughout the system via direct steric interactions, cross-linking proteins, or long-range hydrodynamic interactions, leading to complex spatiotemporal dynamics (Figure 1, middle). These simulations specifically focus on active particles suspended in a viscous fluid leading to orientation-dependent viscosity with significant long-range hydrodynamic and steric interactions.

### 3.1.3 convective_envelope_rsg

Massive stars evolve into red supergiants (RSGs), which have turbulent and convective envelopes. Here, 3D radiative hydrodynamic (RHD) simulations model these convective envelopes, capturing inherently 3D processes like convection (Figure 1, bottom). The simulations give insight into a variety of phenomena: the progenitors of supernovae (SN) explosions and the role of the 3D gas distribution in early SN [58]; calibrations of mixing-length theory (used to model convection in 1D [59–61, 21]); the granulation effects caused by large-scale convective plumes and their impacts on interferometric and photometric observations [62–65].

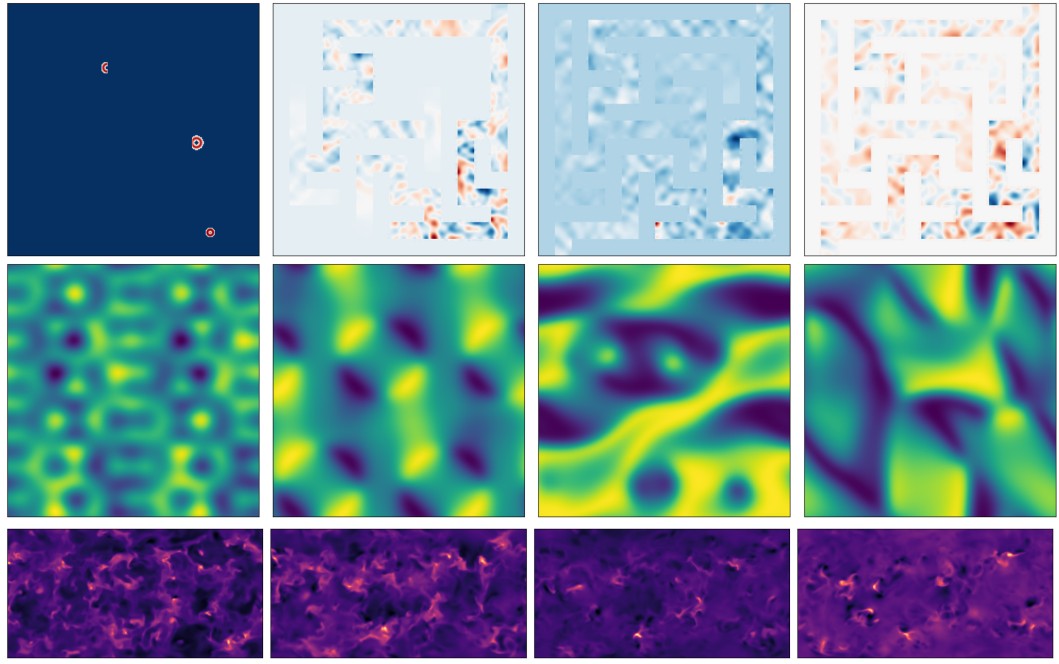

Figure 1: Top to bottom row: snapshots at $t = \{0, \frac{T}{3}, \frac{2T}{3}, T\}$ of `acoustic_scattering`, `active_matter` and `convective_envelope_rsg`.

### 3.1.4  `euler_multi_quadrants`

The Euler equations model the behavior of inviscid fluids. These simulations specifically describe the evolution of compressible gases in a generalization of the classical Euler quadrants Riemann problem [66]. In these problems, initial discontinuities lead to shocks and rarefactions as the system attempts to correct the instability. This dataset is adapted to include multiple initial discontinuities (Figure 2, top) so that the resulting shocks and rarefactions experience further interactions.

### 3.1.5  `gray_scott_reaction_diffusion`

The Gray-Scott model of reaction-diffusion describes the spontaneous assembly of ordered structures from a seemingly disordered system (Figure 2, middle). It occurs across a wide range of biological and chemical systems, often taking place when chemical reactions are coupled to spatial diffusion. For example, reaction–diffusion systems are thought to underpin many of the self-assembly processes present in the early development of organisms [67]. These simulations model the Gray–Scott reaction–diffusion equations [68] describing two chemical species, $A$ and $B$, whose scalar concentrations vary in space and time.

### 3.1.6  `helmholtz_staircase`

Scattering from periodic structures (Figure 2, bottom) occurs in the design of e.g. photonic and phononic crystals, diffraction gratings, antenna arrays, and architecture. These simulations are the linear acoustic scattering of a single point source (which location varies across simulations) from an infinite, periodic, corrugated, sound-hard surface, with unit cells comprising two equal-length line segments.

### 3.1.7  `MHD_64` **and** `MHD_256`

An essential component of the solar wind, galaxy formation, and of interstellar medium (ISM) dynamics is magnetohydrodynamic (MHD) turbulence (Figure 3, top). This dataset consists of isothermal MHD simulations without self-gravity (such as found in the diffuse ISM) initially generated with resolution $256^3$ and then downsampled to $64^3$ after anti-aliasing with an ideal low-pass filter.

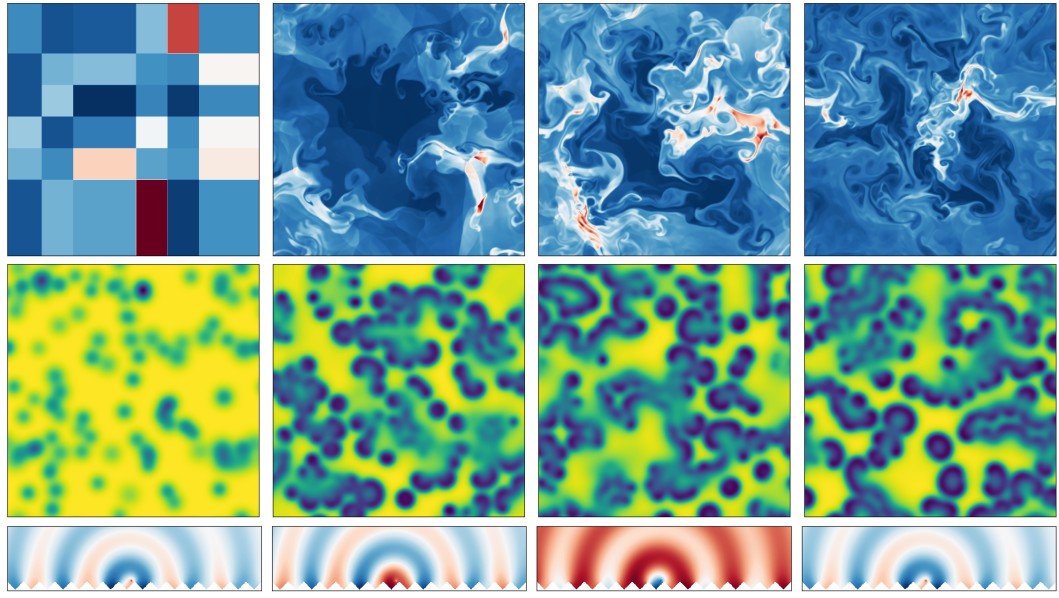

Figure 2: Top to bottom row: snapshots at $t = \{0, \frac{T}{3}, \frac{2T}{3}, T\}$ of `euler_multi_quadrants`, `gray_scott_reaction_diffusion`, and `helmholtz_staircase`

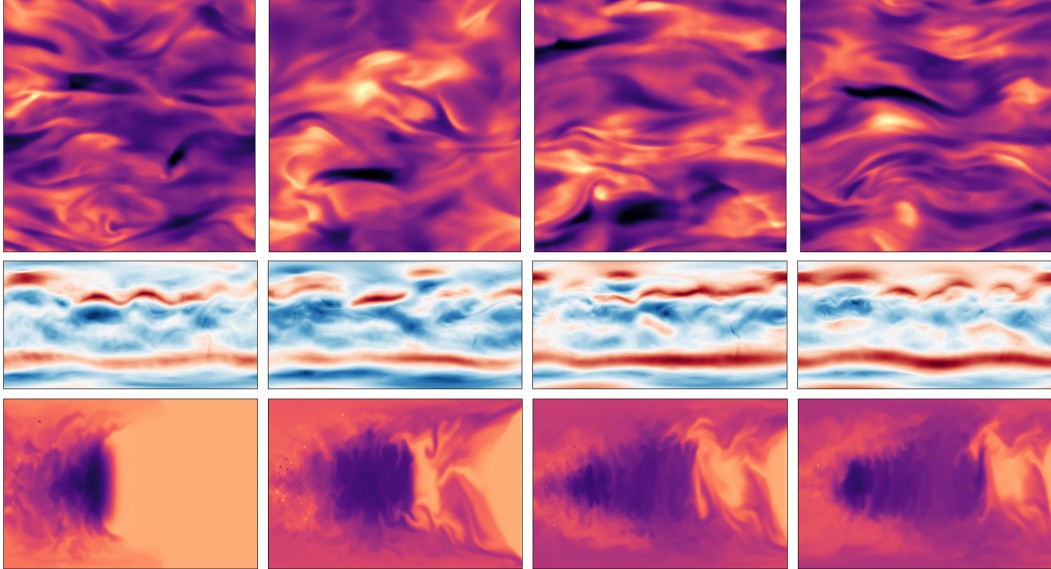

Figure 3: Top to bottom row: snapshots at $t = \{0, \frac{T}{3}, \frac{2T}{3}, T\}$ of `MHD`, `planetswe` and `post_neutron_star_merger`.

### 3.1.8 `planetswe`

The shallow water equations approximate incompressible fluid flows where the horizontal length scale is significantly larger than the vertical as a depth-integrated two-dimensional problem. They have played an important roll in the validation of dynamical cores for atmospheric dynamics as seen in the classical Williamson problems [69]. These simulations can be seen as a refinement of Williamson 7 as they are initialized from the hPa500 level of the ERA5 reanalysis dataset [42] with bathymetry corresponding to the earth's topography and featuring forcings with daily and annual periodicity (Figure 3, middle).

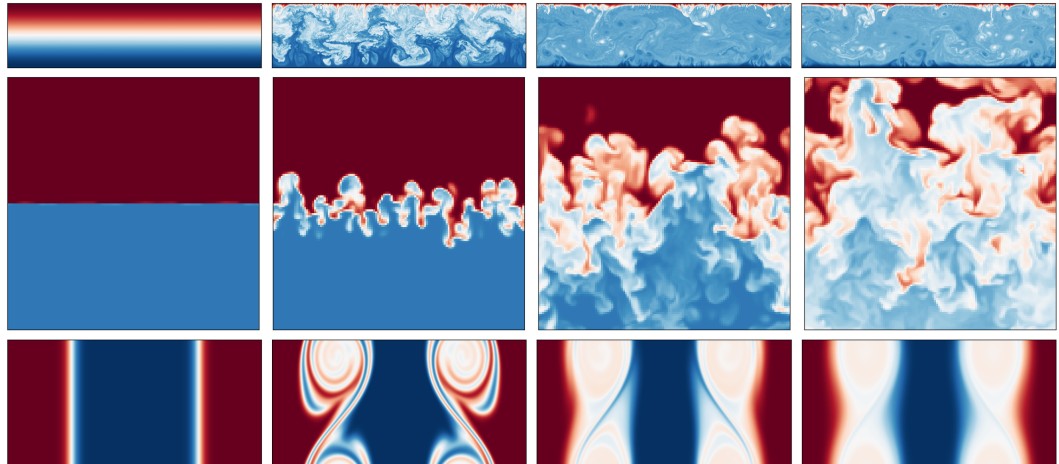

Figure 4: Top to bottom row: snapshots at $t = \{0, \frac{T}{3}, \frac{2T}{3}, T\}$ of `rayleigh_benard`, `rayleigh_taylor_instability` and `shear_flow`.

### 3.1.9 `post_neutron_star_merger`

After the in-spiral and merger of two neutron stars, a hot dense remnant is formed. These events, central to gamma ray bursts and heavy element formation, produce a reddening glow called a *kilonova* [70–77]. Accurate predictions require modeling neutrino interactions, which convert neutrons to protons and vice versa. These simulations model the accretion disk driving the gamma ray burst and the hot neutron-rich wind causing the kilonova (Figure 3, bottom).

### 3.1.10 `rayleigh_benard`

Rayleigh-Bénard convection [78, 79] is a phenomenon in fluid dynamics encountered in geophysics (mantle convection [80], ocean circulation [81], atmospheric dynamics [82]), in engineering (cooling systems [83], material processing [84]), in astrophysics (interior of stars and planets [85]). It occurs in a horizontal layer of fluid heated from below and cooled from above. This temperature difference creates a density gradient that can lead to the formation of convection currents, where warmer, less dense fluid rises, and cooler, denser fluid sinks (Figure 4, top).

### 3.1.11 `rayleigh_taylor_instability`

The Rayleigh-Taylor instability [86] is comprised of two fluids of different densities initially at rest. The instability arises from any perturbation that will displace a parcel of heavier fluid below a parcel of lighter fluid (Figure 4, middle). Pressure forces are then not aligned with density gradients and this generates vorticity, increasing the amplitude of the perturbations. Eventually, these amplitudes become so large that non-linear turbulent mixing develops.

### 3.1.12 `shear_flow`

Shear flow phenomena [87–89] occurs when layers of fluid move parallel to each other at different velocities, creating a velocity gradient perpendicular to the flow direction (Figure 4, bottom). This can lead to various instabilities and turbulence, which are fundamental to many applications in engineering (e.g., aerodynamics [90]), geophysics (e.g., oceanography [91]), and biomedicine (e.g. biomechanics [92]).

### 3.2 `supernova_explosion_64` **and** `supernova_explosion_128`

Supernova explosions happen at the end of the lives of some massive stars. These explosions release high energy into the interstellar medium (ISM) and create blastwaves. The blastwaves accumulate in the ISM and form dense, sharp shells, which quickly cool down and can be new star-forming regions (Figure 5, top). These small explosions have a significant impact on the entire galaxy's evolution.

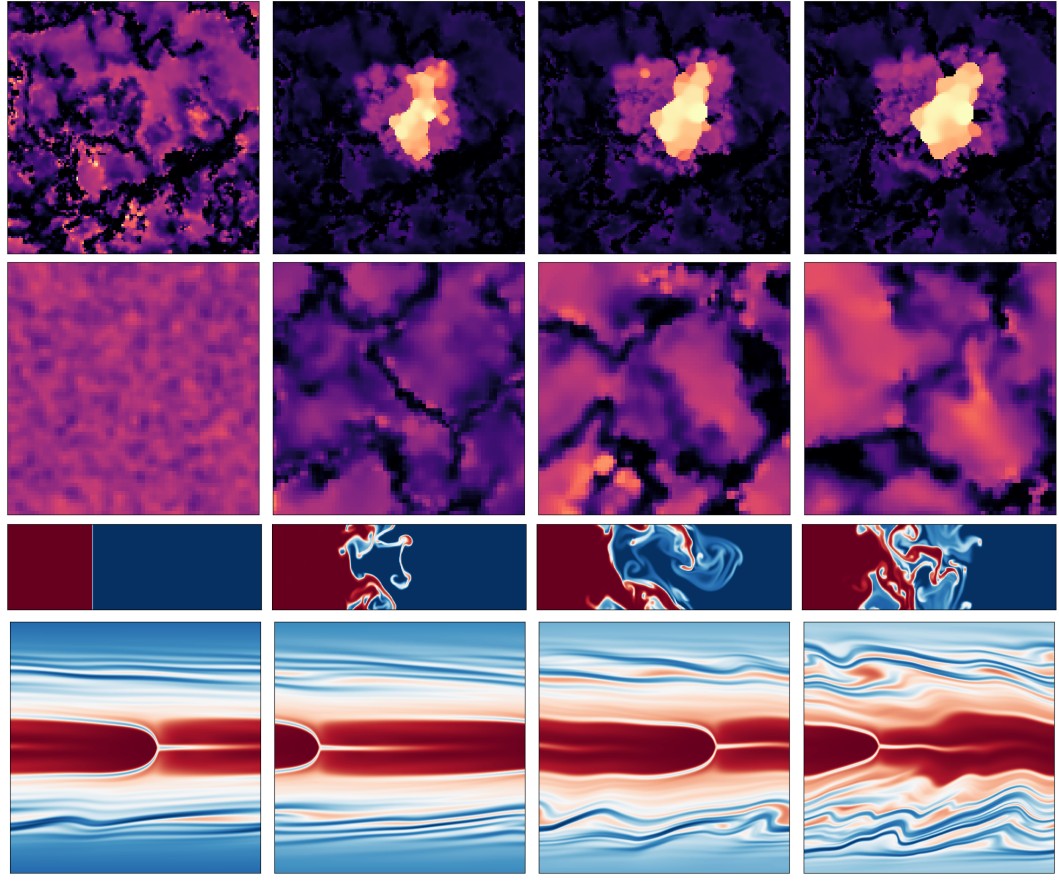

Figure 5: Top to bottom row: snapshots at $t = \{0, \frac{T}{3}, \frac{2T}{3}, T\}$ of `supernova_explosion`, `turbulence_gravity_cooling` `turbulent_radiative_layer_2D` and `viscoelastic_instability`.

### 3.3 `turbulence_gravity_cooling`

Within the interstellar medium (ISM), turbulence, star formation, supernova explosions, radiation, and other complex physics significantly impact galaxy evolution. This ISM is modeled by a turbulent fluid with gravity. These fluids make dense filaments (Figure 5, second row), leading to the formation of new stars. The timescale and frequency of making new filaments vary with the mass and length of the system.

#### 3.3.1 `turbulent_radiative_layer_2D` and `turbulent_radiative_layer_3D`

In astrophysical environments, cold dense gas clumps move through a surrounding hotter gas, mixing due to turbulence at their interface. This mixing creates an intermediate temperature phase that cools rapidly by radiative cooling, causing the mixed gas to join the cold phase as photons escape and energy is lost (Figure 5, third row). Simulations and theories show that if cooling is faster (slower) than mixing, the cold clumps will grow (shrink) [93, 94]. These simulations [95] describe the competition between turbulent mixing and radiative cooling at a mixing layer. These simulations are available in 2D and 3D.

#### 3.3.2 `viscoelastic_instability`

In two-dimensional dilute polymer solutions, the flow exhibits four coexistent attractors: the laminar state, a steady arrowhead regime (SAR), a chaotic arrowhead regime (CAR), and a (recently discovered) chaotic regime of elasto-inertial turbulence (EIT). SAR corresponds to a simple traveling wave, while CAR and EIT are visually similar but differ by a weak polymer arrowhead structure across the mid-plane in CAR. These simulations [96] are snapshots (Figure 5, bottom) of the four attractors and two edge states. Edge

| Dataset | Model | | | |
|---|---|---|---|---|
| | FNO | TFNO | U-net | CNextU-net |
| acoustic_scattering (maze) | 0.5062 | 0.5057 | 0.0351 | **0.0153** |
| active_matter | 0.3691 | 0.3598 | 0.2489 | **0.1034** |
| convective_envelope_rsg | **0.0269** | 0.0283 | 0.0555 | 0.0799 |
| euler_multi_quadrants (periodic b.c.) | 0.4081 | 0.4163 | 0.1834 | **0.1531** |
| gray_scott_reaction_diffusion | **0.1365** | 0.3633 | 0.2252 | 0.1761 |
| helmholtz_staircase | **0.00046** | 0.00346 | 0.01931 | 0.02758 |
| MHD_64 | 0.3605 | 0.3561 | 0.1798 | **0.1633** |
| planetswe | 0.1727 | **0.0853** | 0.3620 | 0.3724 |
| post_neutron_star_merger | 0.3866 | **0.3793** | — | — |
| rayleigh_benard | 0.8395 | **0.6566** | 1.4860 | 0.6699 |
| rayleigh_taylor_instability (At = 0.25) | >10 | >10 | >10 | >10 |
| shear_flow | 1.189 | 1.472 | 3.447 | **0.8080** |
| supernova_explosion_64 | 0.3783 | 0.3785 | **0.3063** | 0.3181 |
| turbulence_gravity_cooling | 0.2429 | 0.2673 | 0.6753 | **0.2096** |
| turbulent_radiative_layer_2D | 0.5001 | 0.5016 | 0.2418 | **0.1956** |
| turbulent_radiative_layer_3D | 0.5278 | 0.5187 | 0.3728 | **0.3667** |
| viscoelastic_instability | 0.7212 | 0.7102 | 0.4185 | **0.2499** |

Table 2: **Model Performance Comparison:** VRMSE metrics on test sets (lower is better). Best results are shown in **bold**. VRMSE is scaled such that predicting the mean value of the target field results in a score of 1.

states exist on the boundary between two basins of attraction and have a single unstable direction, marking the boundary between different flow behaviors.

## 4 Benchmark

To showcase the dataset and the associated benchmarking library, we provide a set of simple baselines time-boxed to 12 hours on a single NVIDIA H100 to demonstrate the effectiveness of naive approaches on these challenging problems and motivate the development of more sophisticated approaches. These baselines are trained on the forward problem - predicting the next snapshot of a given simulation from a short history of 4 time-steps. The models used here are the Fourier Neural Operator [97, FNO], Tucker-Factorized FNO [98, TFNO], U-net [99] and a modernized U-net using ConvNext blocks [100, CNextU-net]. The neural operator models are implemented using `neuralop` [101].

We emphasize that these settings are not selected to explore peak performance of modern machine learning, but rather that they reflect reasonable compute budgets and off-the-shelf choices that might be selected by a domain scientist exploring machine learning for their problems. Therefore we focus on popular models using settings that are either defaults or commonly tuned. Full training and hyperparameter details are included in Appendix E.1.

**Results.** Table 2 reports the one-step Variance Scaled Root Mean Squared Error (VRMSE) – defined in Section E.3 – averaged over all physical fields. Table 3 reports VRMSE averaged over time windows in longer rollouts. We report VRMSE over the more common Normalized RMSE (NRMSE) – also defined in the appendix – as we feel the centered normalization is more appropriate for non-negative fields such as pressure or density whose mean is often bounded away from zero. NRMSE, whose denominator is the 2-norm, down-weights errors with respect to these fields even if they have very little variation. We report evaluation on the test set of each model with hyperparameters performing best on the validation set.

**Analysis.** In the next-step prediction setting, the CNextU-net architecture outperforms the others on 8 of the 17 experiments. However, what is very interesting is that there is a noticeable split between problems which favor spatial domain handling and those which prefer the spectral approach. At the one-step level, 9/17 favor U-net type models while 8 favor spectral models. While in some cases, the results are close, in others, one class of models has a clear advantage. The reason for this is not immediately clear. Boundary conditions would be a natural hypothesis as the boundary condition are handled naively according to model defaults which vary between the U-net and FNO-type models, but there is no clear trend in this direction. This performance gap holds if we instead look at the time-averaged losses for different windows of multi-step autoregressive rollouts in Table 3, though we see notably worse performance overall even

| Dataset | FNO | | TFNO | | U-net | | CNextU-net | |
|---|---|---|---|---|---|---|---|---|
| | 6:12 | 13:30 | 6:12 | 13:30 | 6:12 | 13:30 | 6:12 | 13:30 |
| acoustic_scattering(maze) | 1.06 | 1.72 | 1.13 | 1.23 | **0.56** | 0.92 | 0.78 | 1.13 |
| active_matter | >10 | >10 | 7.52 | 4.72 | 2.53 | 2.62 | **2.11** | 2.71 |
| convective_envelope_rsg | **0.28** | 0.47 | 0.32 | 0.65 | 0.76 | 2.16 | 1.15 | 1.59 |
| euler_multi_quadrants | 1.13 | 1.37 | 1.23 | 1.52 | **1.02** | 1.63 | 4.98 | >10 |
| gray_scott_reaction_diffusion | 0.89 | >10 | 1.54 | >10 | 0.57 | >10 | **0.29** | 7.62 |
| helmholtz_staircase | **0.002** | 0.003 | 0.011 | 0.019 | 0.057 | 0.097 | 0.110 | 0.194 |
| MHD_64 | **1.24** | 1.61 | 1.25 | 1.81 | 1.65 | 4.66 | 1.30 | 2.23 |
| planetswe | 0.81 | 2.96 | **0.29** | 0.55 | 1.18 | 1.92 | 0.42 | 0.52 |
| post_neutron_star_merger | 0.76 | 1.05 | **0.70** | 1.05 | — | — | — | — |
| rayleigh_benard | >10 | >10 | >10 | >10 | >10 | >10 | >10 | >10 |
| rayleigh_taylor_instability | >10 | >10 | **6.72** | >10 | >10 | 2.84 | >10 | 7.43 |
| shear_flow | >10 | >10 | >10 | >10 | >10 | >10 | **2.33** | >10 |
| supernova_explosion_64 | 2.41 | >10 | 1.86 | >10 | **0.94** | 1.69 | 1.12 | 4.55 |
| turbulence_gravity_cooling | 3.55 | 5.63 | 4.49 | 6.95 | 7.14 | 4.15 | **1.30** | 2.09 |
| turbulent_radiative_layer_2D | 1.79 | 3.54 | 6.01 | >10 | 0.66 | 1.04 | **0.54** | 1.01 |
| turbulent_radiative_layer_3D | 0.81 | 0.94 | >10 | >10 | 0.95 | 1.09 | **0.77** | 0.86 |
| viscoelastic_instability | 4.11 | — | 0.93 | — | 0.89 | — | **0.52** | — |

Table 3: **Time-Averaged Losses by Window:** VRMSE metrics on test sets (lower is better) averaged over time windows (6-12) and (13-30). Best results are shown in **bold** for (6-12) and underlined for (13-30). VRMSE is scaled such that predicting the mean value of the target field results in a score of 1.

on these relatively short rollouts indicating the difficulty of performing autoregressive rollouts from one-step training alone. The performance gap between these model classes suggests that one-model-fits-all approaches in this space may be difficult.

Furthermore, there are two observations that may be unexpected to the reader in Table 3. First, loss sometimes decreases in later windows. This can be explained by the problem physics. Dissipative solutions become smoother or better mixed as time progresses and thus easier to predict. Though in the cases where we observe this happening, the normalized loss is typically quite large in either case. The second trend is that losses in Table 2 do not always line up with Table 3. This is due to a difference in experiment setup. Longer rollouts are always initiated from the beginning of the simulation while one-step evaluation occurs on sliding windows sampled from the ground truth. Thus the difficulty of the two settings can vary depending on the behavior of the ground truth physics.

## 4.1 Evaluation Metrics

The benchmark library comes equipped with a variety of metrics to inform architecture design in physically meaningful ways. Often, we are interested in more granular analysis than single-valued metrics. For instance, in Figure 6, we explore `turbulent_radiative_layer_2D` using per-field metrics. We can see that loss varies significantly by field and is concentrated in the pressure (P) field. Similarly, looking at one-step performance, it appears that CNextU-net has a sizable advantage, but when we look at longer time horizons, this advantage quickly dissipates and all models apart from the original U-net become largely interchangeable. The binning of this error over frequency bins provides further insight as we see all models effectively predict low frequency modes in the long run, but high frequency modes diverge more quickly. The full collection of metrics available in the included library is described in Appendix E.3

## 4.2 Moving Beyond the Baselines

The baseline models employed here are powerful but naive models employed en masse without accounting for the specific physical characteristics of the datasets. These are just a starting point for analysis with the Well. Areas for further exploration include:

**Physical constraints.** Conservation laws and boundary conditions are both key physical properties that can often be directly controlled by a model [102–106]. The Well features a variety of conserved quantities and diverse boundary conditions that can vary within a single dataset, making it well-suited to advance such research.

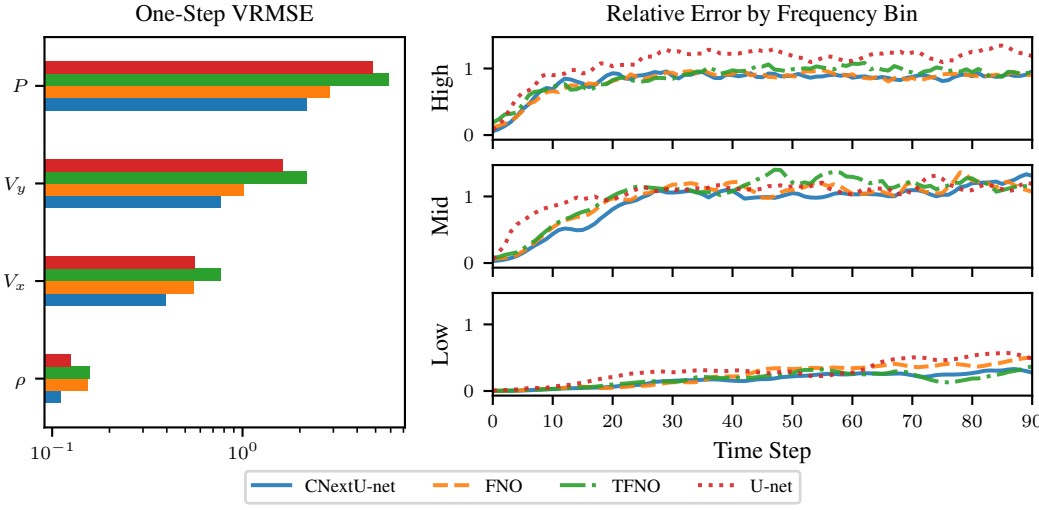

Figure 6: The benchmark library included with the Well includes both coarse and fine level metrics. On the left, we can see the model's performance in VRMSE across state variables. On the right, we divide the isotropic power spectrum into three bins whose boundaries are evenly distributed in log space and evaluate the growth of RMSE per bin normalized by the true bin energy to examine the model's ability to consistently resolve the problem scales.

**Long-term stability.** Several prior studies have highlighted the difficulty and importance of stable surrogate models [20, 49, 107, 108]. The Well is designed with these studies in mind with most datasets including at least 50 snapshots per trajectory while some include a thousand or more.

**Further challenges.** While our example baselines target the forward problem, the Well can be used for a variety of other tasks. Several datasets, such as `acoustic_scattering` and `helmholtz_staircase` are well-suited for inverse scattering tasks. Others like `MHD` are coarsened representations of high resolution simulations and could be used for studies of super-resolution. Many contain wide parameter ranges valuable for generalization studies.

We discuss these and other challenges on a per-dataset basis in Appendix D.

## 5   Conclusion

**Limitations.** These datasets are not without their limitations. They focus largely on uniformly sampled domains at manageable resolutions while many engineering problems require higher resolutions and more complicated meshes than most conventional architectures can feasibly process. These resolution limits often push the use of 2D simulation data while real-world applications are almost always 3D, particularly for turbulent instabilities. Additionally, the Well is primarily a data-focused release. Other works acknowledged in Section 2 explore metrics and analysis more thoroughly. Our focus here is on providing challenging, easily-accessible benchmark problems that can be used in a variety of ways. As available VRAM increases or more efficient architectures are developed, the current version of the Well may no longer be challenging and new datasets may be needed to push the community forward.

Nonetheless, the Well is an important step forward in the development of datasets for physical dynamics prediction. Historically, new challenges have been necessary to push the machine learning community forward. The Well has been developed in collaboration with domain experts to identify problems that provide such unique challenges in more ways than just computational cost. As a collection of datasets containing 15 TB and 16 individual physical scenarios representing physical phenomena of interest to a range of scientific fields, the Well provides the community with a valuable combination of complexity, volume, and diversity that we hope will inspire both new developments in surrogate modeling and perhaps unlock new workflows not yet foreseen.

## Acknowledgments

The authors would like to thank the Scientific Computing Core, a division of the Flatiron Institute, a division of the Simons Foundation, and more specifically Geraud Krawezik for the computing support, the members of Polymathic AI for the insightful discussions, and especially Michael Eickenberg for his input on the paper. Polymathic AI acknowledges funding from the Simons Foundation and Schmidt Sciences, LLC. Additionally, we gratefully acknowledge the support of NVIDIA Corporation for the donation of the DGX Cloud node hours used in this research. The authors would like to thank Aaron Watters, Alex Meng and Lucy Reading-Ikkanda for their help on the visuals, as well as Keaton Burns for his help on using Dedalus. M.B and R.R.K acknowledge Dr Jacob Page and Dr Yves Dubief for their valuable discussions about the multistability of viscoelastic states, and are grateful to EPSRC for supporting this work via grant EP/V027247/1. B.B. acknowledges the generous support of the Flatiron Institute Simons Foundation for hosting the CATS database and the support of NASA award 19-ATP19-0020. R.M. would like to thank Keaton Burns for his advice on using the Dedalus package for generating data. J.S, J.A.G, Y-F J. would like to thank Lars Bildsten, William C. Schultz, and Matteo Cantiello for valuable discussions instrumental to the development of the global RSG simulation setup. These calculations were supported in part by NASA grants ATP-80NSSC18K0560 and ATP-80NSSC22K0725, and computational resources were provided by the NASA High-End Computing (HEC) program through the NASA Advanced Supercomputing (NAS) Division at Ames. J.M.M's work was supported through the Laboratory Directed Research and Development program under project number 20220564ECR at Los Alamos National Laboratory (LANL). LANL is operated by Triad National Security, LLC, for the National Nuclear Security Administration of U.S. Department of Energy (Contract No. 89233218CNA000001). P.M. acknowledges the continued support of the Neutrino Theory Network Program Grant under award number DE-AC02-07CHI11359. P.M. expresses gratitude to the Institute of Astronomy at the University of Cambridge for hosting them as a visiting researcher, during which the idea for this contribution was conceived and initiated. S.S.N would like to acknowledge that their work is funded and supported by the CEA. K.H. acknowledges support of Grants-in-Aid for JSPS Fellows (22KJ1153) and MEXT as "Program for Promoting Researches on the Supercomputer Fugaku" (Structure and Evolution of the Universe Unraveled by Fusion of Simulation and AI; Grant Number JPMXP1020230406). These calculations are partially carried out on Cray XC50 CPU-cluster at the Center for Computational Astrophysics (CfCA) of the National Astronomical Observatory of Japan.

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

# Appendix

## A Datasheet for The Well

### A.1 Motivation

Q1 **For what purpose was the dataset created?** Was there a specific task in mind? Was there a specific gap that needed to be filled? Please provide a description.

- The Well was created to fill two primary gaps. The first is the relationship between diversity and complexity in existing spatiotemporal physics benchmarks. Current products generally either contain a diverse selection of relatively simple physics at low resolution or more in-depth examples of a problem in a single domain. The former are often derived from scenarios used as demonstrations in numerical codes. In these cases, the relative simplicity is understandable since numerical computing and machine learning experts are not fully aware of where the difficulties lay in the other community. Moreover they rarely use the same language. Consequently, when researchers try to demonstrate that new ML architectures are useful across multiple scenarios, it is often on simulations of limited complexity. Integrating the more complex examples requires interfacing with multiple code bases, representation strategies, data types, and formats. This is especially problematic for our second motivation which is the recent movement towards foundation models for spatiotemporal dynamics. In the creation of the Well, we worked with domain experts to find research-frontier level simulations in multiple fields, for which resolution is not the sole challenging factor, that can push the boundaries of generalization of deep learning surrogates and augmented these with more challenging versions of classical scenarios to create a single source with both complexity and diversity.

Q2 **Who created the dataset (e.g., which team, research group) and on behalf of which entity (e.g., company, institution, organization)?**

- These datasets are collected and maintained by Polymathic AI, a research collaboration within the Simons Foundation. Additionally, the Well is a multi-institutional effort that includes researchers from Polymathic, the Flatiron Institute, the University of Cambridge, New York University, Rutgers University, Cornell University, the University of Tokyo, Los Alamos National Laboratory, the University of California Berkeley, Princeton University, and CEA DAM, University of Colorado Boulder and University of Liège.

Q3 **Who funded the creation of the dataset?** If there is an associated grant, please provide the name of the grantor and the grant name and number.

- The project organization was funded by the Simons Foundation, Schmidt Sciences, and received compute donations from the NVIDIA Corporation. Individual researchers contributing datasets were additionally funded by:
  - M.B and R.R.K. - EPSRC grant EP/V027247/1.
  - B.B. - NASA award 19-ATP19-0020
  - J.S., J.A.G., Y-F.J. - NASA grants ATP-80NSSC18K0560, ATP-80NSSC22K0725 with compute support from NASA High-End Computing247 (HEC) program through the NASA Advanced Supercomputing (NAS) Division at Ames.
  - J.M.M - LANL project 20220564ECR. LANL is operated by Triad National Security, LLC, for the National Nuclear Security Administration of U.S. Department of Energy (Contract No.25189233218CNA000001).
  - P.M. - Neutrino Theory Network Program Grant award DE-AC02-07CHI11359.
  - S.S.N - CEA support.
  - K.H. - Grants-in-Aid for JSPS Fellows (22KJ1153) and MEXT as "Program for Promoting Researches on the Supercomputer Fugaku" (Structure and Evolution of the Universe Unraveled by Fusion of Simulation and AI; Grant Number JPMXP1020230406).

Q4 **Any other comments?**

- No.

### A.2 Composition

**Q5 What do the instances that comprise the dataset represent (e.g., documents, photos, people, countries)?** *Are there multiple types of instances (e.g., movies, users, and ratings; people and interactions between them; nodes and edges)? Please provide a description.*

- The Well is a collection of 16 simulation datasets in two and three spatial dimensions totaling 15TB following a common schema and accessible through a unified interface. Each dataset consists of a set of HDF5 files containing snapshots of the physical variables of the simulated system on a discrete grid of spatial and temporal points. Each file is self-documenting containing all field names, dimensions, and simulation parameters in accompanying metadata such that our provided interface can read the metadata and correctly process all included datasets.

**Q6 How many instances are there in total (of each type, if appropriate)?**

- Varies between datasets, but sizes and generation details can be found in Table 4.

**Q7 Does the dataset contain all possible instances or is it a sample (not necessarily random) of instances from a larger set?** *If the dataset is a sample, then what is the larger set? Is the sample representative of the larger set (e.g., geographic coverage)? If so, please describe how this representativeness was validated/verified. If it is not representative of the larger set, please describe why not (e.g., to cover a more diverse range of instances, because instances were withheld or unavailable).*

- It is not sampled from a larger dataset, though the range of physical simulation is significantly larger. The dataset represents a mixture of cutting edge research simulation in various fields and more complex variants of classical phenomena performed at a resolution that is challenging but not insurmountable for current deep learning architectures.

**Q8 What data does each instance consist of?** *"Raw" data (e.g., unprocessed text or images) or features? In either case, please provide a description.*

- An instance can consist of one or more temporal snapshots of the ensemble of state variables in the physical system under simulation. These correspond to rows in the HDF5 `Dataset` objects.

**Q9 Is there a label or target associated with each instance?** *If so, please provide a description.*

- There are not fixed labels, though all datasets in the Well are amenable to temporal rollout tasks where a user predicts future values from historical values. Other datasets are suited to a variety of challenges documented in Appendix C.

**Q10 Is any information missing from individual instances?** *If so, please provide a description, explaining why this information is missing (e.g., because it was unavailable). This does not include intentionally removed information, but might include, e.g., redacted text.*

- No.

**Q11 Are relationships between individual instances made explicit (e.g., users' movie ratings, social network links)?** *If so, please describe how these relationships are made explicit.*

- Yes, temporal and spatial relationships and inherent to the storage format. There is no cross-linkage between different datasets.

**Q12 Are there recommended data splits (e.g., training, development/validation, testing)?** *If so, please provide a description of these splits, explaining the rationale behind them.*

- Yes, within each dataset, for each set of simulation parameters, we apply an 80/10/10 split along the initial conditions. For example, if we have 100 initial conditions for each of 5 simulation parameters and capture 200 steps per simulation, we include 80 of these trajectories of 200 steps per simulation parameter in the training set, 10 in validation, and 10 in test. For datasets with an insufficient number of simulations for this strategy, we apply temporally blocked splitting so that train, validation, and test are large non-overlapping chunks of time with the same split percentage. These are not necessarily in order so that we are not purely testing temporal extrapolation.

**Q13 Are there any errors, sources of noise, or redundancies in the dataset?** *If so, please provide a description.*

- Numerical simulation is not a perfect representation of physical phenomena. As our datasets come from a variety of simulation software using different families of solvers, the solver-specific biases are reduced in our data compared to existing datasets. That said, many of these simulations are under-resolved given the equation parameters used. This lack of resolution can be interpreted as added (numerical) viscosity in the same sense as Implicit Large Eddy Simulation (iLES) models. This is a limitation for parameter inference tasks, but does not effect the majority of use cases.

Q14 **Is the dataset self-contained, or does it link to or otherwise rely on external resources (e.g., websites, tweets, other datasets)?** *If it links to or relies on external resources, a) are there guarantees that they will exist, and remain constant, over time; b) are there official archival versions of the complete dataset (i.e., including the external resources as they existed at the time the dataset was created); c) are there any restrictions (e.g., licenses, fees) associated with any of the external resources that might apply to a future user? Please provide descriptions of all external resources and any restrictions associated with them, as well as links or other access points, as appropriate.*

- The dataset is self-contained.

Q15 **Does the dataset contain data that might be considered confidential (e.g., data that is protected by legal privilege or by doctor–patient confidentiality, data that includes the content of individuals' non-public communications)?** *If so, please provide a description.*

- No, all data was contributed willingly by researchers included in the author list of the paper.

Q16 **Does the dataset contain data that, if viewed directly, might be offensive, insulting, threatening, or might otherwise cause anxiety?** *If so, please describe why.*

- No.

Q17 **Does the dataset relate to people?** *If not, you may skip the remaining questions in this section.*

- No.

Q18 **Does the dataset identify any subpopulations (e.g., by age, gender)?**

- N/A.

Q19 **Is it possible to identify individuals (i.e., one or more natural persons), either directly or indirectly (i.e., in combination with other data) from the dataset?** *If so, please describe how.*

- N/A.

Q20 **Does the dataset contain data that might be considered sensitive in any way (e.g., data that reveals racial or ethnic origins, sexual orientations, religious beliefs, political opinions or union memberships, or locations; financial or health data; biometric or genetic data; forms of government identification, such as social security numbers; criminal history)?** *If so, please provide a description.*

- N/A.

Q21 **Any other comments?**

- No.

### A.3 Collection Process

Q22 **How was the data associated with each instance acquired?** *Was the data directly observable (e.g., raw text, movie ratings), reported by subjects (e.g., survey responses), or indirectly inferred/derived from other data (e.g., part-of-speech tags, model-based guesses for age or language)? If data was reported by subjects or indirectly inferred/derived from other data, was the data validated/verified? If so, please describe how.*

- All data in the Well was produced via numerical simulation.

Q23 **What mechanisms or procedures were used to collect the data (e.g., hardware apparatus or sensor, manual human curation, software program, software API)?** *How were these mechanisms or procedures validated?*

- The collection code varies by dataset. Most datasets were produced from well-maintained open-source numerical software including Clawpack, Dedalus, Athena, and others. These are widely used by researchers and validated extensively within their own projects.

Q24 **If the dataset is a sample from a larger set, what was the sampling strategy (e.g., deterministic, probabilistic with specific sampling probabilities)?**

- N/A.

Q25 **Who was involved in the data collection process (e.g., students, crowdworkers, contractors) and how were they compensated (e.g., how much were crowdworkers paid)?**

- Everyone involved with the generation, collection, and processing of the datasets is included as an author on the paper. The generating software comes from a variety of software projects whose contributors are either users, voluntary contributors, or funded by grants.

Q26 **Over what timeframe was the data collected? Does this timeframe match the creation timeframe of the data associated with the instances (e.g., recent crawl of old news articles)?** *If not, please describe the timeframe in which the data associated with the instances was created.*

- N/A - the materials in the dataset are not date-dependent.

Q27 **Were any ethical review processes conducted (e.g., by an institutional review board)?** *If so, please provide a description of these review processes, including the outcomes, as well as a link or other access point to any supporting documentation.*

- N/A. The data contains no personal information of any kind whether masked or obfuscated or aggregated.

Q28 **Does the dataset relate to people?** *If not, you may skip the remaining questions in this section.*

- No.

Q29 **Any other comments?**

- No.

### A.4 Preprocessing, Cleaning, and/or Labeling

Q30 **Was any preprocessing/cleaning/labeling of the data done (e.g., discretization or bucketing, tokenization, part-of-speech tagging, SIFT feature extraction, removal of instances, processing of missing values)?** *If so, please provide a description. If not, you may skip the remainder of the questions in this section.*

- Before saving simulation results to disk, the datasets are temporally downsampled from their generated resolution - typically the solver will take smaller steps than required to ensure a step ends exactly at the sampling interval. This is done both for storage purposes and to ensure that prediction tasks are non-trivial. The downsampling rates are selected by the domain experts contributing the data with the guidance that the resulting state fields flow smoothly, but where the identity prediction produces non-negligible error.
- Additionally, the file formats are standardized across different datasets. The shared format is documented in Appendix B.3.

Q31 **Was the "raw" data saved in addition to the preprocessed/cleaned/labeled data (e.g., to support unanticipated future uses)?** *If so, please provide a link or other access point to the "raw" data.*

- No, temporal downsampling often occurs by factors upwards of 100 for simulations requiring very small step sizes and saving this to disk is untenable. We do in many instances provide the code to recreate the full data if users wish to save snapshots at different intervals with the caveat that these are complex simulations run on HPC clusters with highly performant code and our generating scripts assume the underlying libraries are already installed and configured for the machine they are running on.

Q32 **Is the software used to preprocess/clean/label the instances available?** *If so, please provide a link or other access point.*

- When possible, we provide the generating code that can be used to recreate the raw data with the caveats mentioned above.

Q33 **Any other comments?**

- No.

### A.5 Uses

Q34 **Has the dataset been used for any tasks already?** *If so, please provide a description.*

- We provide a variety of autoregressive forecasting benchmarks in the submission itself. Several datasets are tied to earlier work, both in machine learning and otherwise, which are mentioned in Appendix C.

Q35 **Is there a repository that links to any or all papers or systems that use the dataset?** *If so, please provide a link or other access point.*

- Yes, this is currently contained in Appendix C and the Github README and we plan to continue updating the README as outside users begin to use the dataset.

Q36 **What (other) tasks could the dataset be used for?**

- Several of the datasets are well suited for other conventional tasks in AI for Science, including inverse acoustic scattering, superresolution, stability challenges, and more documented per-dataset in Section C.

Q37 **Is there anything about the composition of the dataset or the way it was collected and preprocessed/cleaned/labeled that might impact future uses?** *For example, is there anything that a future user might need to know to avoid uses that could result in unfair treatment of individuals or groups (e.g., stereotyping, quality of service issues) or other undesirable harms (e.g., financial harms, legal risks) If so, please provide a description. Is there anything a future user could do to mitigate these undesirable harms?*

- No human applications, but task considerations are discussed in response to the next question.

Q38 **Are there tasks for which the dataset should not be used?** *If so, please provide a description.*

- As many simulations are not fully resolved, we would not recommend this dataset for the evaluation of inverse parameter estimation such as predicting the simulation viscosity from sequences of snapshots. We believe this is true of most existing datasets in this space as fully resolving interesting flows via DNS is computationally onerous.

Q39 **Any other comments?**

- No.

### A.6 Distribution

Q40 **Will the dataset be distributed to third parties outside of the entity (e.g., company, institution, organization) on behalf of which the dataset was created?** *If so, please provide a description.*

- Yes, the dataset will be provided completely openly with contribution guidelines for new contributors.

Q41 **How will the dataset be distributed (e.g., tarball on website, API, GitHub)?** *Does the dataset have a digital object identifier (DOI)?*

- The size of the data makes traditional distribution complicated, as a result the data is hosted by the Flatiron Institute and available either for direct download using provided code or via a Globus endpoint. As this is a collection of individual datasets, they can be used individually or as a group without needing to download the unused datasets. The dataset is too large for direct hosting by a service like Zenodo, but we plan on linking the Github repository to Zenodo upon full release and obtain a DOI. We also plan to distribute the datasets via HuggingFace.

Q42 **When will the dataset be distributed?**

- Download information is available at the Github repository `https://github.com/PolymathicAI/the_well`.

Q43 **Will the dataset be distributed under a copyright or other intellectual property (IP) license, and/or under applicable terms of use (ToU)?** *If so, please describe this license and/or ToU, and provide a link or other access point to, or otherwise reproduce, any relevant licensing terms or ToU, as well as any fees associated with these restrictions.*

- CC-BY-4.0

**Q44 Have any third parties imposed IP-based or other restrictions on the data associated with the instances?** *If so, please describe these restrictions, and provide a link or other access point to, or otherwise reproduce, any relevant licensing terms, as well as any fees associated with these restrictions.*

- All available generating code is similarly provided under CC-BY-4.0, though some datasets were generated with proprietary code that cannot be released.

**Q45 Do any export controls or other regulatory restrictions apply to the dataset or to individual instances?** *If so, please describe these restrictions, and provide a link or other access point to, or otherwise reproduce, any supporting documentation.*

- No.

**Q46 Any other comments?**

- No.

### A.7 Maintenance

**Q47 Who will be supporting/hosting/maintaining the dataset?**

- The Flatiron Institute will host and maintain the data and Globus endpoint going forward.
- We plan for datasets to become available via HuggingFace.

**Q48 How can the owner/curator/manager of the dataset be contacted (e.g., email address)?**

- Our preferred strategy is via issues on the Github page `https://github.com/PolymathicAI/the_well/issues`. Corresponding authors may also be contacted directly, though the Github is recommended as individual contributors may leave or join the collaboration over time.

**Q49 Is there an erratum?** *If so, please provide a link or other access point.*

- There is no erratum for our initial release. Errata will be documented as future releases on the dataset website.

**Q50 Will the dataset be updated (e.g., to correct labeling errors, add new instances, delete instances)?** *If so, please describe how often, by whom, and how updates will be communicated to users (e.g., mailing list, GitHub)?*

- Our Github page contains contributor guidelines that may be used to add additional datasets to the Well.

**Q51 If the dataset relates to people, are there applicable limits on the retention of the data associated with the instances (e.g., were individuals in question told that their data would be retained for a fixed period of time and then deleted)?** *If so, please describe these limits and explain how they will be enforced.*

- N/A. No people.

**Q52 Will older versions of the dataset continue to be supported/hosted/maintained?** *If so, please describe how. If not, please describe how its obsolescence will be communicated to users.*

- Yes. As the Well is a collection of datasets, existing datasets will be maintained and any modification to the collection will be in the form of added datasets, so data will not become out-of-date.

**Q53 If others want to extend/augment/build on/contribute to the dataset, is there a mechanism for them to do so?** *If so, please provide a description. Will these contributions be validated/verified? If so, please describe how. If not, why not? Is there a process for communicating/distributing these contributions to other users? If so, please provide a description.*

- Yes, contributor guidelines are included on the Github, though it is not an automated process. Contribution requires working with the team at Polymathic AI on validating and formatting the data and ensuring that the size of the contributed data is manageable with the existing distribution strategy. New data will also have to undergo preliminary benchmarking to ensure that it is compatible with ML workflows.
- For users who wish to use their own data in Well-based workflows, our provided interface includes the tools to do so.

Q54 **Any other comments?**

- No.

# B  How to build The Well

## B.1  Initial Construction

The Well was built using the following organization method:

- Domain scientists and numerical software developers were contacted. Individuals working with simulations that were sufficiently distinct from existing datasets, non-trivial for learning, and did not require excessive resolution were brought into the collaboration.

- Domain experts were asked to generate data across a sensible range of simulation parameters or initial conditions given the complexity of their simulations. They generated the data on the clusters associated with their home institution.

- The data was then transferred to the Flatiron Institute cluster for storage and processing.

- Data was analyzed to ensure there were no `NaN`, that the grid and time steps were uniform, and that the data files were consistent.

- A data specification was created for storage, distribution, and programmatic access for machine learning users.

- The data was processed into this common format. A PyTorch Dataset was constructed to read this data for machine learning usage.

- Once processed, a compute budget was allocated to benchmarking based on the size of the data and typical workloads in the space.

- Preliminary benchmarking was performed and results were reported in the paper.

## B.2  Data Availability

The Well is hosted by the Flatiron Institute which has hosted a number of large datasets over a sustained period of time. We are in discussion for making subsets available on HuggingFace upon release. During the review process, code for downloading the data can be found at the following repository.

## B.3  Data Specification

We provide the data with a unified data specification and PyTorch-based interface. The data resides in HDF5 archives with a shared format.

The specification is described below with example entries for a hypothetical 2D ($D=2$) simulation with dimension B x T x W x H. Note that this uses HDF5 Groups, Datasets, and attributes (denoted by "@"):

```
root: Group
  @simulation_parameters: list[str] = ['ParamA', ...]
  @ParamA: float = 1.0
  ... # Additional listed parameters
  @dataset_name: str = 'ExampleDSet'
  @grid_type:
    str = 'cartesian' # "cartesian/spherical currently supported"
  @n_spatial_dims:
    int = 2 # Should match number of provided spatial dimensions.
  @n_trajectories: int = B # "Batch" dimension of dataset

  -dimensions: Group
    @spatial_dims:
    list[str] = ['x', 'y'] # Names match datasets below.
    time: Dataset = float32(T)
      @sample_varying
    = False # Does this value vary between trajectories?
    -x: Dataset = float32(W) # Grid coordinates in x
```

```
      @sample_varying = False
      @time_varying = False # True not currently supported
    -y = float32(H) # Grid coordinates in y
      @sample_varying = False
      @time_varying = False

  -boundary_conditions:
    Group # Internal and external boundary conditions
     -X_boundary: Group
       @associated_dims: list[str] = ['x'] # Defined on x
       # If associated with set values for given field.
       @associated_fields: list[str] = []
       #
    Geometric description of BC. Currently support periodic/wall/open
       @bc_type = 'periodic'
       @sample_varying = False
       @time_varying = False
       -mask:
    Dataset = bool(W) # True on coordinates where boundary is defined.
       -values:
    Dataset = float32(NumTrue(mask)) # Values defined on mask points

  scalars: Group # Non-spatially varying scalars.
    @field_names: list[str] = ['ParamA', 'OtherScalar', ...]
    ParamA: Dataset = float32(1)
      @sample_varying = False # Does this vary between trajectories?
      @time_varying = False # Does this vary over time?
    OtherScalar: Dataset = float32(T)
      @sample_varying = False
      @time_varying = True

  t0_fields: Group
    # field_names should list all datasets in this category
    @field_names: list[str] = ['FieldA', 'FieldB', 'FieldC', ...]
    -FieldA: Dataset = float32(BxTxWxH)
      @dim_varying = [ True   True]
      @sample_varying = True
      @time_varying = True
    -FieldB: Dataset = float32(TxWxH)
      @dim_varying = [ True   True]
      @sample_varying = True
      @time_varying = False
    -FieldC: Dataset = float32(BxTxH)
      @dim_varying = [ True   False]
      @sample_varying = True
      @time_varying = True
    ... # Additional fields

  -t1_fields: Group
    @field_names = ['VFieldA', ...]
    -VFieldA: Dataset = float32(BxTxWxHxD)
      @dim_varying = [ True   True]
      @sample_varying = True
      @time_varying = True
    ... # Additional fields

  -t2_fields: Group
    @field_names: list[str] = ['TFieldA', ...]
    - TFieldA: Dataset = float32(BxTxWxHxD^2)
      @antisymmetric = False
      @dim_varying = [ True   True]
      @sample_varying = True
      @symmetric = True # Whether tensor is symmetric
      @time_varying = True
    ... # Additional fields
```

We did not generate Croissant [109] descriptions of the datasets because the specification does not currently support HDF5 files and would have required converting the 15TB of data to another format that is handled by the standard. Our provided specification is self-documenting and contains sufficient metadata for machine processing.

For usage purposes, the `GenericWellDataset` outputs all fields as a dictionary giving users the option of how to arrange the input and output for their goals. We include default data processors which add all time-invariant fields as model inputs, but not as targets.

| Dataset | Size (GB) | Run time (h) | Hardware | Software |
|---|---|---|---|---|
| acoustic_scattering_discontinuous | 157 | 0.25 | 64 C | Clawpack [110] |
| acoustic_scattering_inclusions | 283 | 0.25 | 64 C | Clawpack [110] |
| acoustic_scattering_maze | 311 | 0.33 | 64 C | Clawpack [110] |
| active_matter | 51.3 | 0.33 | A100 GPU | Python |
| convective_envelope_rsg | 570 | 1460 | 80 C | Athena++ [111] |
| euler_multi_quadrants | 5170 | 80* | 160 C* | ClawPack [110] |
| helmholtz_staircase | 52 | 0.11 | 64 C | Python |
| MHD_256 | 4580 | 48 | 64 C | Fortran MPI |
| MHD_64 | 72 | – | – | – |
| gray_scott_reaction_diffusion | 154 | 33* | 40 C | Matlab |
| planetswe | 186 | 0.75 | 64 C | Dedalus [112] |
| post_neutron_star_merger | 110 | 505* | 300 C* | $\nu$bhlight [113] |
| rayleigh_benard | 358 | 60* | 768 C* | Dedalus [112] |
| rayleigh_taylor_instability | 256 | 65* | 128 C* | TurMix3D [114] |
| shear_flow | 547 | 5 | 96 C | Dedalus [112] |
| supernova_explosion_128 | 754 | 4* | 1040 C* | ASURA-FDPS [115] |
| supernova_explosion_64 | 268 | 4* | 1040 C* | ASURA-FDPS [115] |
| turbulence_gravity_cooling | 829 | 577* | 1040* C | ASURA-FDPS [115] |
| turbulent_radiative_layer_2D | 6.9 | 2* | 48 C | Athena++ [111] |
| turbulent_radiative_layer_3D | 745 | 271* | 128 C | Athena++ [111] |
| viscoelastic_instability | 66 | 34* | 64 C | Dedalus [112] |

Table 4: Information about the different dataset generation. In the running time and hardware columns, * denotes a total for all the runs. Otherwise, these figures are given for running one simulation only. For hardware, C denotes the number of Cores. Computation was performed on nodes equipped with either 2 48-core AMD Genoa or 2 32-core Intel Icelake.

## C   Dataset Details

All numerical simulations are on a uniform grid, uniform time-steps and in single precision `fp32`.

### C.1   `acoustic_scattering`

**Description of the physical phenomenon.** We include three variants of an acoustic scattering problem to showcase the challenges introduces by sharp discontinuities and irregular structure. The acoustic equations describe the evolution of an acoustic pressure wave through materials with spatially varying density. The specific modeling equations used here are:

$$\frac{\partial p}{\partial t} + K(x,y)\left(\frac{\partial u}{\partial x} + \frac{\partial v}{\partial y}\right) = 0 \tag{1}$$

$$\frac{\partial u}{\partial t} + \frac{1}{\rho(x,y)}\frac{\partial p}{\partial x} = 0 \tag{2}$$

$$\frac{\partial v}{\partial t} + \frac{1}{\rho(x,y)}\frac{\partial p}{\partial v} = 0 \tag{3}$$

with $\rho$ the material density, $u,v$ the velocity in the $x,y$ directions respectively, $p$ the pressure, and $K$ the bulk modulus. $\rho$ and $K$ jointly define the speed of sound and so only $\rho$ is varied in these simulations while $K$ is maintained at a constant value of $4$.

These equations are most prevalent in inverse problems like source optimization of a signal or inverse scattering where the underlying material densities are inferred from observed dynamics. These are simple linear dynamics, but the sharp discontinuities in the underlying material density lead to interesting behavior that can be challenging for learned models.

The three datasets vary in the families of material density configurations they consider:

- *Single Discontinuity* - The simplest example consisting of two continuously varying subdomains with a discontinuous interface. The initial conditions consist of a flat pressure static field with 1-4 high pressure rings randomly placed in domain. The rings are defined with variable intensity $\sim \mathcal{U}(.5,2)$ and radius $\sim \mathcal{U}(.06,.15)$. The subdomain densities are generated from one of the following randomly selected functions:
    - Gaussian Bump - Peak density samples from $\sim \mathcal{U}(1,7)$ and $\sigma \sim \mathcal{U}(.1,5)$ with the center of the bump uniformly sampled from the extent of the subdomain.
    - Linear gradient - Four corners sampled with $\rho \sim \mathcal{U}(1,7)$. Inner density is bilinearly interpolated.
    - Constant - Constant $\rho \sim \mathcal{U}(1,7)$
    - Smoothed Gaussian Noise - Constant background sampled $\rho \sim \mathcal{U}(1,7)$ with i.i.d. standard normal noise applied. This is then smoothed by a Gaussian filter of varying sigma $\sigma \sim \mathcal{U}(5,10)$
- *Inclusions* - In this dataset, we first generate a background from the single discontinuity set and further add randomly generated potentially overlapping "inclusions" containing wildly different material properties. This is akin to a geoscience setting with interfaces and mineral deposits. The inclusions are added as 1-15 random ellipsoids with center uniformly sampled from the domain and height/width sampled uniformly from [.05, .6]. The ellipsoid is then rotated randomly with angle sampled [-45, 45]. For the inclusions, $Ln(\rho) \sim \mathcal{U}(-1,10)$.
- *Maze* - This dataset explores complex arrangements of sharp discontinuities. We generated a maze with initial width between 6 and 16 pixels and upsample it via nearest neighbor resampling to create a 256 x 256 maze. The walls are set to $\rho = 10^6$ while paths are set to $\rho = 3$. The initial sources are generated as a flat pressure static field with 1-6 high pressure rings randomly placed along paths of maze. The rings are defined with variable intensity $\sim \mathcal{U}(3.,5.)$ and radius $\sim \mathcal{U}(.01,.04)$. Any overlap with walls is removed.

**Simulation details.** The simulations are performed using the total variation diminishing solvers in Clawpack [110], a framework for solving hyperbolic conservation laws using an explicit finite volume scheme, with a monotonized central-difference flux limiter with step-size determined by the CFL condition. The simulation occurs on a domain that is open in the $y$ direction and closed (reflective) in the $x$ direction. Each simulation took approximately 15 minutes of wall time on 64 Icelake CPU cores. Parallelization is done using domain decomposition with ghost node padding for internal boundaries. As the maze simulations are run for more steps, they each required 20 minutes.

**Varied Physical Parameters.** We vary $\rho$ while keeping $K$ constant to control the material speed of sound $c$.

**Fields present in the data.** $\boldsymbol{u}$ or $u,v$ the vector-valued velocity field, $p$ the pressure, and constant fields $\rho$ and $c$ (the material speed of sound).

**References to cite when using these simulations:** [110].

### C.2 `active_matter`

**Description of the physical phenomenon.** We are interested in studying the dynamics of $N$ active particles of length $\ell$ and thickness $b$ (aspect ratio $\ell/b \gg 1$) immersed in a Stokes fluid with cubic volume $V$. In large particle limit, continuum kinetic theories describing the evolution of the distribution function $\Psi(\mathbf{x},\mathbf{p},t)$ have proven to be useful tools for analyzing and simulating particle suspensions [116, 117]. The Smoluchowski equation governs $\Psi$'s evolution, ensuring particle number conservation,

$$\frac{\partial \Psi}{\partial t} + \nabla_{\mathbf{x}} \cdot (\dot{\mathbf{x}}\Psi) + \nabla_{\mathbf{p}} \cdot (\dot{\mathbf{p}}\Psi) = 0, \tag{4}$$

where the conformational fluxes $\dot{\mathbf{x}}$ and $\dot{\mathbf{p}}$ are obtained from the dynamics of a single particle in a background flow $\mathbf{u}(\mathbf{x},t)$. The moments of $\Psi$ yield the concentration field $c = \langle 1 \rangle$, polarity field $\mathbf{n} = \langle \mathbf{p} \rangle / c$,

and nematic order parameter $\mathbf{Q} = \langle\mathbf{pp}\rangle/c$, with $\langle f \rangle = \int_{|\mathbf{p}|=1} f\Psi \, d\mathbf{p}$. For dense suspensions, the conformational fluxes are

$$\dot{\mathbf{x}} = \mathbf{u} - d_T \nabla_{\mathbf{x}} \log\Psi; \quad \dot{\mathbf{p}} = (\mathbf{I} - \mathbf{pp}) \cdot (\nabla\mathbf{u} + 2\zeta\mathbf{D}) \cdot \mathbf{p} - d_R \nabla_{\mathbf{p}} \log\Psi. \tag{5}$$

Here $d_T$ and $d_R$ are dimensionless translational and rotational diffusion constants, $\zeta$ is the strength of particle alignment through steric interactions, and $\mathbf{D} = \langle\mathbf{pp}\rangle$ is the second-moment tensor. The Smoluchowski equation is coupled to the Stokes flow as

$$-\Delta\mathbf{u} + \nabla P = \nabla\cdot\mathbf{\Sigma}, \nabla\cdot\mathbf{u} = 0, \tag{6}$$

$$\mathbf{\Sigma} = \alpha\mathbf{D} + \beta\mathbf{S}:\mathbf{E} - 2\zeta\beta(\mathbf{D}\cdot\mathbf{D} - \mathbf{S}:\mathbf{D}). \tag{7}$$

Here $P(\mathbf{x},t)$ is the fluid pressure, $\alpha$ is the dimensionless active dipole strength, $\beta$ characterizes the particle density, $\mathbf{E} = [\nabla\mathbf{u} + \nabla\mathbf{u}^\top]/2$ is the symmetric rate-of-strain tensor, and $\mathbf{S} = \langle\mathbf{pppp}\rangle$ is the fourth-moment tensor. The stress tensor $\mathbf{\Sigma}$ in Eq. (7) includes contributions from active dipole strength, particle rigidity, and local steric torques. Despite the fact that kinetic theories are consistent with microscopic details and are amenable to analytical treatment, they are not immune from computational challenges. For instance, in dense suspensions with strong alignment interactions (high $\zeta$), the cost to resolve the orientation field $\mathbf{p}$ is prohibitively high even in 2D. Though approximate coarse-grained models that track only low-order moments exist, they rely on phenomenological [118][119] or learned corrections [120] to close the system. This underscores the need for fast, high-fidelity, data-efficient physical surrogate models to track and predict the evolution of few low-order moments. An autoregressive surrogate model can efficiently screen the high-dimensional parameter space of complex active matter systems and help design self-organizing materials that switch between nontrivial dynamical states in response to external actuation or varying parameters.

**Simulation details.** We numerically close the system of equations (4)-(6) using pseudo-spectral discretization where Fourier differentiation is used to evaluate the derivatives with respect to space and particle orientation. We use the second order implicit-explicit backward differentiation time-stepping scheme (SBDF2), where the linear terms are handled implicitly and the nonlinear terms explicitly with time-step $\Delta t = 0.0004$. The numerical simulations are performed in a periodic square domain of length $L = 10$ with $256^2$ spatial modes and 256 orientational modes. The simulation code is available at `https://github.com/SuryanarayanaMK/Learning_closures/tree/master`. The approximate time to generate the data is 20 minutes per simulation on an A100 80GB GPU in `fp64` precision. In total, this is about 75 hours of simulation.

**Varied Physical Parameters.** $\alpha \in \{-1,-2,-3,-4,-5\}$ $\beta = 0.8$; $\zeta \in \{1,3,5,7,9,11,13,15,17\}$.

**Fields present in the data.** concentration (scalar field), velocity (vector field), orientation tensor (tensor field), strain-rate tensor (tensor field).

**References to cite when using these simulations:** [120].

## C.3 `convective_envelope_rsg`

**Description of the physical phenomenon.** The 3D radiation hydrodynamic (RHD) equations are [121]:

$$\frac{\partial\rho}{\partial t} + \boldsymbol{\nabla}\cdot(\rho\boldsymbol{v}) = 0 \tag{8}$$

$$\frac{\partial(\rho\boldsymbol{v})}{\partial t} + \boldsymbol{\nabla}\cdot(\rho\boldsymbol{v}\boldsymbol{v} + \mathsf{P}_{\text{gas}}) = -\mathbf{G}_r - \rho\boldsymbol{\nabla}\Phi \tag{9}$$

$$\frac{\partial E}{\partial t} + \boldsymbol{\nabla}\cdot[(E + P_{\text{gas}})\boldsymbol{v}] = -cG_r^0 - \rho\boldsymbol{v}\cdot\boldsymbol{\nabla}\Phi \tag{10}$$

$$\frac{\partial I}{\partial t} + c\boldsymbol{n}\cdot\boldsymbol{\nabla}I = S(I,\boldsymbol{n}) \tag{11}$$

where $\rho$ is the gas density, $\boldsymbol{v}$ is the flow velocity, $\mathsf{P}_{\text{gas}}$ and $P_{\text{gas}}$ are the gas pressure tensor and scalar, respectively, $E$ is the total gas energy density, with $E = E_g + \rho v^2/2$, where $E_g = 3P_{\text{gas}}/2$ is the gas internal energy density, $G_r^0$ and $\mathbf{G}_r$ are the time-like and space-like components of the radiation four-force, and $I$ is the frequency integrated intensity, which is a function of time, spatial coordinate, and photon propagation direction $\boldsymbol{n}$. $\boldsymbol{\nabla}\Phi$ is defined as $\boldsymbol{\nabla}\Phi = -Gm(r)/r^2$, where $m(r)$ is the mass inside the radial coordinate $r$ including the mass contained within the simulation inner boundary. The source term describing the interaction between the gas and radiation in a co-moving frame as given by

$$S_0(I_0,\boldsymbol{n_0}) = c\rho\kappa_{aP}\left(\frac{ca_r T^4}{4\pi} - J_0\right) + c\rho(\kappa_S + \kappa_{aR})(J_0 - I_0), \tag{12}$$

where $\kappa_{aP}$ and $\kappa_{aR}$ are Planck and Rosseland mean absorption opacities from OPAL [122], and and $\kappa_S$ is the electron scattering opacity, all evaluated in the co-moving frame. These simulations neglect stellar rotation and magnetic fields. Similar setups have been used by [123, 124].

**Simulation details.** The RHD equations are solved using the standard Godunov method in `Athena++` [111], available at `https://www.athena-astro.app/`. The simulation grid is in spherical-polar coordinates with 128 uniform bins in polar angle $\theta$ from $\frac{\pi}{4} - \frac{3\pi}{4}$ and 256 bins in azimuth $f$ from $0 - \pi$ with periodic boundary conditions in $\theta$ and $f$. Outside of the simulation domain, `Athena++` uses ghost zones to enforce its boundary conditions. For the "periodic" boundary in $\theta$, the ghost zones from $\pi/4$ ($3\pi/4$) are copied from the last active zones around the $3\pi/4$ ($\pi/4$) boundary so that the mass and energy flux across the $\theta$ boundary is conserved. The radial direction is covered by a logarithmic spaced grid consisting of 384 (256) zones, with $\delta r/r \approx 0.01$, extending far out enough to capture any wind structure or extended atmosphere. The simulations were generated during 2 months on 80 nodes of NASA PLeiades Skylake CPU nodes.

**Varied Physical Parameters.** All simulations are cuts of a larger simulation. They have all the same physical parameters, but are different times of the same simulation.

**Fields present in the data.** : energy (scalar field), density (scalar field), pressure (scalar field), velocity (vector field).

**References to cite when using these simulations:** [21].

### C.4 `euler_multi_quadrants`

**Description of the physical phenomenon.** This particular set of simulations solves the compressible inviscid Euler equations, which in two dimensions in integral form are

$$\frac{d}{dt}\iint_\Omega U dA + \oint_{\partial\Omega} (F\hat{i} + G\hat{j})\cdot\hat{n}dS = 0 \tag{13}$$

where $U = (\rho, \rho u, \rho v, \rho E)^T$ and

$$F = \begin{pmatrix} \rho u \\ \rho u^2 + p \\ \rho uv \\ u(\rho E + p) \end{pmatrix} \quad G = \begin{pmatrix} \rho v \\ \rho uv \\ \rho v^2 + p \\ v(\rho E + p) \end{pmatrix}. \tag{14}$$

Here, $\rho$ is the density, $u$ and $v$ are the Cartesian velocities, $p$ is the pressure, and $\rho E = p/(\gamma - 1) + \frac{1}{2}\rho(u^2 + v^2)$ is the total velocity.

**Simulation details.** These simulations used the open source software `CLAWPack` [110, 125], a general framework for solving hyperbolic conservation laws using an explicit finite volume scheme. The simulations use different sets of piecewise constant initial data, which is known as a Riemann problem [66]. The possible solutions are then a combination of shocks, rarefaction waves, or contact discontinuities that sometimes interact as the simulation proceeds in time. The data was generated in `fp64` in 80 hours on 160 CPU cores.

**Varied Physical Parameters.** $\gamma \in \{1.3, 1.4, 1.13, 1.22, 1.33, 1.76, 1.365, 1.404, 1.453, 1.597\}$ and boundary conditions are either open or periodic.

**Fields present in the data.** density (scalar field), energy (scalar field), pressure (scalar field), momentum (vector field).

**References to cite when using these simulations:** [110, 125].

### C.5 `gray_scott_reaction_diffusion`

**Description of the physical phenomenon.** The Gray–Scott equations [68] are a set of coupled reaction–diffusion equations describing two chemical species, $A$ and $B$, whose scalar concentrations vary in space and time:

$$\frac{\partial A}{\partial t} = \delta_A \Delta A - AB^2 + f(1 - A), \tag{15}$$

$$\frac{\partial B}{\partial t} = \delta_B \Delta B + AB^2 - (f + k)B. \tag{16}$$

The two parameters $f$ and $k$ control the "feed" and "kill" rates in the reaction, respectively; specifically, $f$ controls the rate at which species $A$ is added to the system and $k$ controls the rate at which species $B$ is removed. The two diffusion constants $\delta_A$ and $\delta_B$ govern the rate of diffusion of each species. A zoo of qualitatively different static and dynamic patterns in the solutions are possible depending on the two parameters $f$ and $k$ [126]. There is a rich landscape of pattern formation hidden in these equations.

**Simulation details.** Many numerical methods exist to simulate reaction–diffusion equations. If low-order finite differences are used, real-time simulations can be carried out using GPUs, with modern browser-based implementations readily available [126, 127]. We choose to simulate with a high-order spectral method here for accuracy and stability purposes. We simulate (15)–(16) in two dimensions on the doubly periodic domain $[-1,1]^2$ using a Fourier spectral method implemented in the MATLAB package Chebfun [128]. Specifically, we use the implicit-explicit exponential time-differencing fourth-order Runge–Kutta method [129] to integrate this stiff PDE in time. The Fourier spectral method is used in space, with the linear diffusion terms treated implicitly and the nonlinear reaction terms treated explicitly and evaluated pseudospectrally. Simulations are performed using a $128 \times 128$ bivariate Fourier series over a time interval of 10,000 seconds, with a simulation time step size of 1 second. Snapshots are recorded every 10 time steps. We seed the simulation trajectories with 200 different initial conditions: 100 random Fourier series and 100 randomly placed Gaussians. In all simulations, we set $\delta_A = 0.00002$ and $\delta_B = 0.00001$. Pattern formation is then controlled by the choice of the "feed" and "kill" parameters $f$ and $k$. We choose six different $(f,k)$ pairs which result in six qualitatively different patterns, summarized in the following table.

|         | $f$   | $k$   |
|---------|-------|-------|
| Gliders | 0.014 | 0.054 |
| Bubbles | 0.098 | 0.057 |
| Maze    | 0.029 | 0.057 |
| Worms   | 0.058 | 0.065 |
| Spirals | 0.018 | 0.051 |
| Spots   | 0.030 | 0.062 |

On 40 CPU cores, it takes 5.5 hours per set of parameters, 33 hours in total for all simulations.

**Varied Physical Parameters.** All simulations used $\delta_u = 2.10^{-5}$ and $\delta_v = 1.10^{-5}$. "Gliders": $f = 0.014$, $k = 0.054$. "Bubbles": $f = 0.098$, $k = 0.057$. "Maze": $f = 0.029$, $k = 0.057$. "Worms": $f = 0.058$, $k = 0.065$. "Spirals": $f = 0.018$, $k = 0.051$. "Spots": $f = 0.03$, $k = 0.062$.

**Fields present in the data.** Two chemical species $A$ and $B$.

**References to cite when using these simulations:** None.

### C.6 `helmholtz_staircase`

**Description of the physical phenomenon.** We simulate linear acoustic scattering of a single point source from an infinite, periodic, corrugated, sound-hard surface. The region $\Omega \in \mathbb{R}^2$ above the boundary $\partial\Omega$ is simply connected and filled with a constant-density gas with sound speed $c > 0$. We define $\mathbf{x} = (x_1, x_2)$. The boundary $\partial\Omega$ extends with spatial period $d$ in the $x_1$ direction and unbounded in the perpendicular $x_2$ direction. The current example is a right-angled staircase whose unit cell consists of two equal-length line segments at $\pi/2$ angle to each other, see Fig. 1 in [130]. This geometry models a 3D staircase which extends infinitely in the third direction pointing into the plane of the paper. While we solve the problem in the frequency domain, the original time-domain problem is described by the wave equation sourced by a point excitation at $t = 0$ and $\mathbf{x} = \mathbf{x}_0 \in \Omega$,

$$\frac{\partial^2 U(t,\mathbf{x})}{\partial t^2} - \Delta U(t,\mathbf{x}) = \delta(t)\delta(\mathbf{x}-\mathbf{x}_0) \qquad t \in \mathbb{R}, \quad \mathbf{x} \in \Omega, \tag{17}$$

where $\Delta = \nabla \cdot \nabla$ is the spatial Laplacian, and time $t$ is rescaled such that the sound speed $c = 1$. We assume quiescence before the point excitation: $U \equiv 0$ for $t < 0$, and that the normal component of the fluid velocity vanishes at the staircase's surface, yielding Neumann boundary conditions

$$U_n(t,\mathbf{x}) = \mathbf{n} \cdot \nabla U(t,\mathbf{x}) = 0 \qquad t \in \mathbb{R}, \quad \mathbf{x} \in \partial\Omega, \tag{18}$$

where $\mathbf{n}$ is the unit boundary normal pointing into $\Omega$. Taking the Fourier transform with respect to $t$ of Eqs. (17)–(18), we get the inhomogeneous Helmholtz Neumann boundary value problem (BVP) that is

the focus of this simulation,

$$-(\Delta + \omega^2)u = \delta_{\mathbf{x}_0} \quad \text{in } \Omega, \tag{19}$$

$$u_n = 0 \quad \text{on } \partial\Omega, \tag{20}$$

where $\omega \in \mathbb{R}$ is the emission frequency of the source. We solve for the acoustic pressure $u$, which is additionally subject to radiation conditions as described in [130].

Scattering from periodic structures occurs in real-life applications such as the design of waveguides on various lengthscales: photonic and phononic crystals, diffraction gratings, antenna arrays, and architectural elements. These applications often involve numerical simulations performed repeatedly in an optimization or inference loop, calling for fast and robust numerical methods. This setting, however, presents some challenges to accurate numerical modeling. The solution domain is unbounded in both the vertical direction and along the surface; truncation in the vertical direction requires satisfying the correct radiation conditions, and naive truncation in the horizontal direction would result in large artificial reflections (and hence errors) due to the possibility of waves being guided along the surface. Periodization—reducing the computation to the unit cell—is seemingly impossible, since the point source breaks the periodicity of the problem. It is possible, however, to express the nonperiodic solution in terms of a family of quasiperiodic solutions via the Floquet–Bloch transform (also referred to as the array scanning method). The current geometric setup involves corner singularities that must be dealt with if high-order accuracy is to be achieved. Finally, as the input frequency $\omega$ grows, the computation will become more expensive due to the need for a finer discretization grid to resolve oscillations.

**Simulation details.** Our simulation combines the Floquet–Bloch transform with a high-order boundary integral equation (BIE) method to solve each of the quasiperiodic BVPs. The main advantage of this approach is a reduction of the number of discretization nodes (and hence computational cost) by conversion of the 2D PDE to an integral to be evaluated on a 1D boundary. High-order accuracy is then achievable via appropriately chosen quadrature rules, which can easily handle the corner singularities. In contrast, finite difference (FD) and finite element (FEM) schemes require finer meshing of the domain near the corners and implement radiation conditions explicitly. The Floquet–Bloch transform has previously been paired with both FD and FEM methods to tackle scattering from a nonperiodic source, but only to low-order accuracy [131, 132]. Other approaches include meshfree methods such as the method of fundamental solutions [133–135] and the plane waves method [136], as well as tools based on the Rayleigh hypothesis [137]. In the high-frequency limit, fast methods exist that exploit approximations including the Helmholtz–Kirchhoff approximation [138] and geometric acoustics [139–141].

The Helmholtz staircase dataset consists of 25600 images generated from 512 distinct input parameter combinations; the parameters are the source frequency $\omega$ (takes 16 different values) and the source position $\mathbf{x}_0$ (takes 32 values). All input frequencies lie in the "low-frequency" regime in the sense that there exists a trapped acoustic mode at that frequency, meaning that the input wavelength is of the same lengthscale as the staircase period. For each parameter combination, we generate 50 timesteps spanning one temporal period, $T = 2\pi/\omega$, analytically via $U(t,x) = u(t,x)\exp(-i\omega t)$. The simulations are accurate to around 7–8 digits.

We chose the low-frequency regime for the purposes of training due to the existence of trapped modes in this limit. One can identify two distinct spatial frequencies in the generated images: the input frequency $\omega$, which dominates the outgoing waves far away from the boundary, and the *distinct* spatial frequency of the trapped mode visible along the boundary. The prediction algorithm needs to learn that out of the two, it is $\omega$ that determines the time-dependence of the acoustic waves, and correctly identify it from the image despite the presence of a trapped mode. This gets increasingly difficult as $\omega$ rises (due to the two frequencies growing more disparate), until a cutoff above which trapped modes no longer exist. In the future, it would be of interest to also learn the dispersion relation of trapped modes, i.e. infer the relationship between their spatial frequency and the input frequency based on the boundary geometry. The code to generate the simulations will be available at `https://github.com/fruzsinaagocs/periodic-bie`. On 64 CPU cores, the simulation takes $\sim 400s$ per input parameter, total $\sim 50$ hours.

**Varied Physical Parameters.** frequency of the source $\omega \in \{0.062, 0.251, 0.439, 0.626, 0.813, 0.998, 1.182, 1.363, 1.541, 1.715, 1.882, 2.042, 2.191, 2.323, 2.433, 2.511\}$, with the sources coordinates being all combinations of $x \in \{-0.4, -0.3, -0.2, -0.1, 0, 0.1, 0.2, 0.3, 0.4\}$ and $y \in \{-0.2, -0.1, 0, 0.1, 0.2, 0.3, 0.4\}$.

**Fields present in the data.** real and imaginary part of acoustic pressure (scalar field), the staircase mask (scalar field, stationary).

**References to cite when using these simulations:** [130].

## C.7 `MHD` (magnetohydrodynamic simulations)

**Description of the physical phenomenon.** These simulations employ a third-order-accurate hybrid essentially non-oscillatory (ENO) scheme [142] to solve the ideal MHD equations:

$$\frac{\partial \rho}{\partial t} + \nabla \cdot (\rho \boldsymbol{v}) = 0, \tag{21}$$

$$\frac{\partial \rho \boldsymbol{v}}{\partial t} + \nabla \cdot \left[ \rho \boldsymbol{v}\boldsymbol{v} + \left( p + \frac{B^2}{8\pi} \right) \mathbf{I} - \frac{1}{4\pi} \mathbf{BB} \right] = \mathbf{f}, \tag{22}$$

$$\frac{\partial \mathbf{B}}{\partial t} - \nabla \times (\boldsymbol{v} \times \mathbf{B}) = 0. \tag{23}$$

Here, $\rho$ is the density, $v$ the velocity, $\mathbf{B}$ denotes the magnetic field, $p$ represents the gas pressure, and $\mathbf{I}$ is the identity matrix. These simulations utilize periodic boundary conditions and an isothermal equation of state, $p = c_s^2 \rho$, where $c_s$ is the isothermal sound speed. For the energy source term $\mathbf{f}$, we assume a random large-scale solenoidal driving at a wave number $k \approx 2.5$ (i.e., 1/2.5 of the box size), with continuous driving. The simulations are executed with a $256^3$ grid resolution and have been referenced and utilized in numerous prior studies [142–146].

The main control parameters of these MHD simulations are the dimensionless sonic Mach number, $\mathcal{M}s \equiv |\boldsymbol{v}|/cs$, and the Alfvénic Mach number, $\mathcal{M}_A \equiv |\boldsymbol{v}|/\langle v_A \rangle$, where $\boldsymbol{v}$ is the velocity, $cs$ and $v_A$ are the isothermal sound speed and the Alfvén speed respectively, and $\langle \cdot \rangle$ signifies averages over the entire simulation box. A range of sonic Mach numbers is provided for two different regimes of Alfvénic Mach number (see below varied physical parameters). The simulations are either sub-Alfvénic with $\mathcal{M}_A \approx 0.7$ (indicating a strong magnetic field) or super-Alfvénic with $\mathcal{M}_A = 2.0$. The initial Alfvén Mach number in the super-Alfvénic runs is 7.0, but after the small-scale dynamo saturates, the final $\mathcal{M}_A$ value is around 2. These simulations are non-self-gravitating, and the file units are in code units. The MHD simulations are scale-free, allowing users to assign a physical scale to the box length and density [147, 148]. Rescaling these simulations requires maintaining the sonic and Alfvén Mach numbers constant, though other physical quantities (e.g., density, velocity) may be converted to physical units. On 64 CPU cores, it takes 48 hours per simulation.

**Varied Physical Parameters.** dimensionless sonic Mach number $\mathcal{M}s \in \{0.5, 0.7, 1.5, 2.0, 7.0\}$ and dimensionless Alfvénic Mach number $\mathcal{M}_A \in \{0.7, 2.0\}$.

**Fields present in the data.** Density (scalar field), velocity (vector field), magnetic field (vector field).

**References to cite when using these simulations:** [142, 149, 146, 150].

## C.8 `planetswe`

**Description of the physical phenomenon.** The shallow water equations are a 2D approximation of a 3D flow in the case where horizontal length scales are significantly longer than vertical length scales. They are derived from depth-integrating the incompressible Navier-Stokes equations. The integrated dimension then only remains in the equation as a variable describing the height of the pressure surface above the flow. In this case, we specifically explore the rotating forced hyperviscous spherical shallow water equations defined as:

$$\frac{\partial \boldsymbol{u}}{\partial t} = -\boldsymbol{u} \cdot \nabla \boldsymbol{u} - g\nabla h - \nu \nabla^4 \boldsymbol{u} - 2\Omega \times \boldsymbol{u} \tag{24}$$

$$\frac{\partial h}{\partial t} = -H\nabla \cdot \boldsymbol{u} - \nabla \cdot (h\boldsymbol{u}) - \nu \nabla^4 h + F \tag{25}$$

where $\nabla^4$ denotes a hyperviscosity term. Hyperviscosity is largely non-physical but is commonly used in atmospheric modeling to maintain stability of under-resolved simulations without effecting large scales to the same degree as conventional diffusion. $\nu = 1.76 \times 10^{-10}$ is therefore selected for simulation stability - equivalently to matching at wave number $224$. $F$ is a forcing term designed to introduce seasonality.

These equations have long been used as a simpler approximation of the primitive equations in atmospheric modeling of a single pressure level, most notably in the Williamson test problems. The scenario in this dataset can be seen as similar to Williamson Problem 7 as we derive initial conditions from the hPa 500 pressure level in ERA5. These are then simulated with realistic topography and two levels of periodicity. Since this is supposed to present a simplified version of the challenges in atmospheric prediction, $F$ is constructed to be a time-dependent forcing term with annual and daily seasonality giving the simulation

a sense of "days" and "years", though these are defined in simulation time rather than in physical units. The logic for $F$ is defined in code as:

```python
def find_center(t):
    time_of_day = t / day
    time_of_year = t / year
    max_declination = .4
    lon_center = time_of_day*2*np.pi
    lat_center = np.sin(time_of_year*2*np.pi)*max_declination
    lon_anti = np.pi + lon_center
    return lon_center, lat_center, lon_anti, lat_center

def season_day_forcing(phi, theta, t, h_f0):
    phi_c, theta_c, phi_a, theta_a = find_center(t)
    sigma = np.pi/2
    coefficients = np.cos(phi - phi_c) \
        * np.exp(-(theta-theta_c)**2 / sigma**2)
    forcing = h_f0 * coefficients
    return forcing
```

**Simulation details.** The simulations are performed using the spin-spherical harmonic pseudospectral method in Dedalus [112] with initial conditions derived from the $u,v,z$ fields in the hPa 500 level of ERA5 [42]. The spatial grid is oversampled by a factor of 3/2 relative to the spectral grid as an anti-aliasing measure following Orszag's rule. To ensure stable initialization, these unbalanced initial conditions are repeatedly simulated for short sequences then projected into hydrostatic balance. The resulting initial conditions are then burned-in for half a model year. The next three model years are then recorded at an interval of one model hour resulting in a total of 3024 recorded steps per initial condition. The simulation time-step varies according to the CFL condition and is performed using a second-order IMEX Runge-Kutta scheme [151]. The resulting data was interpolated onto a equiangular grid by resampling from the spectral representation.

Each simulation took approximately 45 minutes of wall time on 64 Icelake CPU cores.

**Varied Physical Parameters.** This data varies only in initial conditions as it is intended to roughly approximate the challenges associated with a specific physical object (the earth).

**Fields present in the data.** $u$ or $u,v$ the vector-valued velocity field and $h$ the surface height.

**References to cite when using these simulations:** [152]

### C.9 `post_neutron_star_merger`

**Description of the physical phenomenon.** These simulations are of the disk of hot, dense gas formed after the in-spiral and merger of two neutron stars. These cataclysmic events are now known to be the central engines of gamma ray bursts—some of the most energetic events in the universe—and a primary source of heavy elements in the universe [70–72]. The radioactive decay of heavy elements fused in these systems produces a reddening glow that can be seen from earth, a *kilonova,* the first observation of which was made in 2017 [73–77]. Of key importance in predicting these events is capturing accurately the interaction of neutrinos, subatomic particles that interact with nuclei and nucleons to convert neutrons to protons and vice versa. The models here are the most advanced simulations in the world of the accretion disk that drives the relativistic jet that powers the gamma ray burst and of the hot neutron-rich wind that sources one component of the kilonova. We solve the general relativistic equations of ideal magnetohydrodynamics coupled to lepton conservation and the neutrino transport equation:[2]

$$\partial_t\big(\sqrt{g}\rho_0 u^t\big)+\partial_i\big(\sqrt{g}\rho_0 u^i\big) \;=\; 0 \tag{26}$$

$$\partial_t\big[\sqrt{g}\big(T^t_\nu+\rho_0 u^t\delta^t_\nu\big)\big]+\partial_i\big[\sqrt{g}\big(T^i_\nu+\rho_0 u^i\delta^t_\nu\big)\big] \;=\; \sqrt{g}\big(T^\kappa_\lambda\Gamma^\lambda_{\nu\kappa}+G_\nu\big) \;\forall\nu=0,1,...,4 \tag{27}$$

$$\partial_t\big(\sqrt{g}B^i\big)+\partial_j\big[\sqrt{g}\big(b^j u^i-b^i u^j\big)\big] \;=\; 0 \tag{28}$$

$$\partial_t\big(\sqrt{g}\rho_0 Y_e u^t\big)+\partial_i\big(\sqrt{g}\rho_0 Y_e u^i\big) \;=\; \sqrt{g}G_{\text{ye}} \tag{29}$$

$$\frac{D}{d\lambda}\left(\frac{h^3\mathcal{I}_{\nu,f}}{\varepsilon^3}\right) \;=\; \left(\frac{h^2\eta_{\nu,f}}{\varepsilon^2}\right)-\left(\frac{\varepsilon\chi_{\nu,f}}{h}\right)\left(\frac{h^3\mathcal{I}_{\nu,f}}{\varepsilon^3}\right), \tag{30}$$

---

[2]Below we use Einstein summation notation. Repeated indices are summed. Greek indices range from 0 to 3 inclusive. Latin indices range from 1 to 3 inclusive.

where here $\rho_0$ is the rest mass density, $g$ is the absolute value of the determinant of the metric tensor, $u^\mu$ the is the fluid four-vector, $T^\mu_\nu$, the stress energy tensor, $\delta^\mu_\nu$, the Kronecker delta, $\Gamma^\lambda_{\mu\nu}$ the Christoffel symbols, $B^i$ the magnetic field 3-vector, $b^\mu$ the magnetic field four-vector, $Y_e$ the electron fraction (ratio of electrons to baryons, $G_\nu$ the radiation field four-force, $G_{ye}$ the lepton exchange source term. $\mathcal{I}_{\nu,f}$ is the neutrino intensity as a function of position, energy $\varepsilon = h\nu$, and flavor $f$. $d/d\lambda$ is the total derivative along null geodesics of $\mathcal{I}$. $\eta_{\nu,f}$ is the emissivity and $\chi_{\nu,f}$ the opacity. $h$ is Planck's constant.

Roughly, equation (26) is conservation of mass or particle number. Equation (27) is conservation of energy and momentum. Conservation of momentum is of course Newton's second law, but in general relativity this is combined with conservation of energy. Equation (28) is conservation of magnetic flux. In ideal hydrodynamics, conductivities are assumed to be infinite and thus electric fields can be ignored. Magnetic field lines then get advected with the fluid flow. Equation (29) is conservation of lepton number and controls how neutrons and electrons are advected with the fluid. Equation (30) evolves the motion of neutrinos, which are binned into three flavors: electron neutrinos, their antiparticles, and "heavy neutrinos" which include muon and tau neutrinos and their antiparticles. We assume neutrino mass is negligible and approximate the neutrinos as traveling at the speed of light. Thus we are able to use the radiative transfer equation for photons with some modification. For more details, see [113]. The simulations provided in the Well are from a series of papers, [153–156].

These simulations are computationally expensive and challenging. They require sufficiently high resolution and short time scales to capture the magnetorotational instability, which drives fluid motion [157–159]. But they must also be run for sufficiently long times to track the motion of outgoing material. The electron fraction, $Y_e$ is a critical parameter for heavy element nucleosynthesis which ultimately determines the kilonova signal. ML algorithm that captures bulk fluid motion and tracks the electron fraction $Y_e$ without requiring detailed modeling of magnetohydrodynamic turbulence would be a powerful tool in modeling these systems.

**Simulation details.** These simulations were produced using the open source $\nu$bhlight code, available at `https://github.com/lanl/nubhlight` and first described in [113]. This code builds on a long history of methods spanning more than two decades [160–163]. It solves the equations of ideal general relativistic magnetohydrodynamics via finite volume methods with constrained transport, and uses Monte Carlo methods to perform neutrino radiation transport. The two are coupled via first-order operator splitting. The code uses a radially logarithmic quasi-spherical grid in horizon penetrating coordinates, as first described in [164], the WENO reconstruction first described in [165], the primitive variable recovery scheme described in [166], and the drift-frame artificial atmosphere treatment described in [167].

Simulations were generated using the `torus_cbc` problem generator, which constructs a torus of gas in hydrostatic equilibrium around a rotating black hole, as first detailed in [168, 169]. Initial conditions must specify a black hole mass and and angular momentum, an initial disk entropy, electron fraction, inner radius and radius of maximum pressure, and the preferred units of density (usually chosen so that the peak density is close to 1 in code units). A ratio of gas pressure to magnetic pressure at the point of maximum pressure must also be chosen. (This parameter is called plasma $\beta$.) Parameters to reproduce can be found in the cited papers. Finally a finite temperature nuclear equation of state and neutrino opacities must be chosen. The equation of state is the SFHo [170] model. The opacities are the `Fornax` opacities [171] first described in [172]. Both opacities and equation of state are tabulated in Stellar Collapse format [173] and may be found on the web at `https://stellarcollapse.org/`. Each simulation takes $\sim 3$ weeks to be generated using 300 CPU cores.

**Varied Physical Parameters.** Black hole spin parameter a, ranges 0 to 1. Initial mass and angular momentum of torus. In dimensionless units, evaluated as inner radius $R_{in}$ and radius of maximum pressure $R_{max}$. Torus initial electron fraction Ye and entropy kb. Black hole mass in solar masses.

**Fields present in the data.** fluid density (scalar field), fluid internal energy (scalar field), electron fraction (scalar field), temperate (scalar field), entropy (scalar field), velocity (vector field), magnetic field (vector field), contravariant tensor metric of space-time (tensor field, no time-dependency).

**References to cite when using these simulations:** [153–156].

## C.10  `rayleigh_benard`

**Description of the physical phenomenon.** We consider a 2D horizontally-periodic fluid. We write $u = (u_x, u_z)$ its velocity (horizontal and vertical), $b$ its buoyancy which is the upward force exerted on the fluid due to differences in density, themselves caused by difference in temperature, and $p$ the pressure.

With the lower plate heated and the upper cooled, thermal energy creates density variations, initiating fluid motion. This results in Bénard cells, showcasing warm fluid rising and cool fluid descending, which position is highly sensitive to initial conditions. The fluid is governed by the equations:

$$\frac{\partial b}{\partial t} - \kappa \Delta b = -u \cdot \nabla b,$$

$$\frac{\partial u}{\partial t} - \nu \Delta u + \nabla p - b\boldsymbol{e}_z = -u \cdot \nabla u,$$

where $\Delta = \nabla \cdot \nabla$ is the spatial Laplacian and $\boldsymbol{e}_z$ is the unit vector in the vertical direction, with the additional constraint $\int p = 0$ (pressure gauge). The first equation rules the convection and diffusion in the fluid, while the second equation is a Navier-Stokes equation augmented by the buoyancy force. The fluid is periodic in the horizontal direction but it has boundary conditions in the vertical direction at the bottom $z = 0$ and at the top $z = Lz$ as follows $u(z=0)=0$, $b(z=0)=Lz$ and $u(Lz=0)=0$, $b(Lz=0)=0$.

The fluid equations are parameterized by the Rayleigh and Prandtl numbers through the thermal diffusivity $\kappa$ and viscosity $\nu$

$$\kappa = \left(\text{Rayleigh} \times \text{Prandtl}\right)^{-\frac{1}{2}},$$

$$\nu = \left(\frac{\text{Rayleigh}}{\text{Prandtl}}\right)^{-\frac{1}{2}}.$$

The Rayleigh number is a dimensionless parameter that measures the relative importance between the effect of the buoyancy forces and the effect of the viscosity forces and thermal conduction. The Prandtl number is a dimensionless parameter that measures the relative importance between momentum diffusivity and thermal diffusivity [79].

**Simulation details.** The data is generated by solving the PDEs through spectral methods using the Dedalus software [112]. The solution is evolved over time with adaptive time-steps. High Rayleigh simulations are very time-consuming because they require very small time-step to prevent the solution from diverging [174]. The simulation takes between $\sim 6000s$ and $\sim 50000s$ (high Rayleigh number simulations take longer to be generated), 60h in total for all simulations.

**Varied Physical Parameters.** Rayleigh $\in \{1e6, \ 1e7, \ 1e8, \ 1e9, \ 1e10\}$, Prandtl $\in \{0.1, 0.2, 0.5, 1.0, 2.0, 5.0, 10.0\}$. For initial conditions $\delta b_0 \in \{0.2, 0.4, 0.6, 0.8, 1.0\}$.

**Fields present in the data.** buoyancy (scalar field), pressure (scalar field), velocity (vector field).

**References to cite when using these simulations:** [112]

### C.11 rayleigh_taylor_instability

**Description of the physical phenomenon.** The key dimensionless parameter for RTI is the dimensionless density difference or Atwood number ($A = (\rho_h - \rho_l)/(\rho_h + \rho_l)$). As RTI is found to be self-similar, the growth rate ($\alpha$) of the mixing can be characterized by

$$\alpha = \frac{\dot{L}^2}{4AgL}, \tag{31}$$

where $L$ is the width of the turbulent mixing zone.

The flow is governed by equations for continuity, momentum and incompressibility in the case of miscible fluids with common molecular diffusivity:

$$\partial_t \rho + \boldsymbol{\nabla} \cdot (\rho \boldsymbol{u}) = 0, \tag{32}$$

$$\partial_t (\rho \boldsymbol{u}) + \boldsymbol{\nabla} \cdot (\rho \boldsymbol{u} \boldsymbol{u}) = -\boldsymbol{\nabla} p + \boldsymbol{\nabla} \cdot \boldsymbol{\tau} + \rho \boldsymbol{g}, \tag{33}$$

$$\boldsymbol{\nabla} \cdot \boldsymbol{u} = -\kappa \boldsymbol{\nabla} \cdot \left(\frac{\boldsymbol{\nabla} \rho}{\rho}\right). \tag{34}$$

Here, $\rho$ is density, $\boldsymbol{u}$ is velocity, $p$ is pressure, $\boldsymbol{g}$ is gravity, $\kappa$ is the coefficient of molecular diffusivity and $\boldsymbol{\tau}$ is the deviatoric stress tensor

$$\boldsymbol{\tau} = \rho \nu \left(\boldsymbol{\nabla} \boldsymbol{u} + (\boldsymbol{\nabla} \boldsymbol{u})^T - \frac{2}{3}(\boldsymbol{\nabla} \cdot \boldsymbol{u})\boldsymbol{I}\right), \tag{35}$$

where $\nu$ is the kinematic viscosity and $\boldsymbol{I}$ is the identity matrix.

From a fundamental standpoint, we would expect a good machine learning-based model or emulator to advect and mix the density field rather than create or destroy mass to give appropriate statistics. Our simulations are of comparable spatial resolution to simulations run by a large-scale study of the growth rate of RTI [175]. Therefore, we would consider a good emulator to produce a comparable value for the growth rate as reported in their paper for an appropriately similar set of initial conditions. In addition, during the non-linear regime, as turbulence develops, we would expect to observe typical energy spectra of the inertial cascade where energy is distributed following an appropriate $k^{-5/3}$ slope.

From a structural perspective, we would expect that for an initialization with a large variety of modes in the initial spectrum to observe a range of bubbles and spikes (upward and downward moving structures). In the other limit (where there is only one mode in the initial spectrum) we would hope to observe a single bubble and spike [176]. Finally, a good emulator would exhibit a statistically symmetric mixing width for low Atwood numbers in the Boussinesq regime (defined as $A < 0.1$ [177]) and asymmetries in the mixing width for large Atwood number.

**Simulation details.** We use TURMIX3D [114] to solve the governing equations (32), (33) and (34) on a staggered 'Marker and Cell' type mesh [178] using a 'Lagrange + remap' method with a Helmholtz–Hodge type decomposition. The domain is discretized such that each cell is a cube (i.e. $\Delta x = \Delta y = \Delta z = h$) and parallelized in all three directions using MPI.

The code is second-order in space using an upwind total variation diminishing approach with Van Leer flux limiters [179, 180] and second-order in time using a strong stabilization preserved Runge-Kutta [181]. Our discretized pressure equation is modified to account for the non-zero divergence of velocity fields and large density difference and reads as

$$\boldsymbol{\nabla} \cdot \left[ \frac{1}{\rho^{n+1}} \boldsymbol{\nabla} \left( \frac{\rho_l}{\rho_h - \rho_l} p^n \right) \right] = \frac{\rho_l}{\Delta t(\rho_h - \rho_l)} \boldsymbol{\nabla} \cdot \left( \frac{(\rho^{int} \boldsymbol{u}^{int})}{\rho^{n+1}} + \kappa \frac{\boldsymbol{\nabla} \rho^{n+1}}{\rho^{n+1}} \right), \tag{36}$$

where indices $n$ and $n+1$ refer to times $t^n$ and $t^{n+1}$ and the index $int$ refers to an intermediate time incorporating all remaining forces of the momentum equation. Equation (36) is then solved using a 'red and black' relaxation method coupled with a 'V-cycle' multigrid convergence method [182–184]. The coefficient $\rho_l/(\rho_h - \rho_l)$ normalizes the diffusion term to make the pressure solver quasi-independent of the Atwood number [114]. Finally, we must comment on the treatment of viscosity in the code. The kinematic viscosity, $\nu$, is re-scaled to keep the Kolmogorov scale

$$\eta = \nu^{3/4} \langle \varepsilon \rangle^{-(1/4)}, \tag{37}$$

on the order of the mesh resolution. Here $\langle \varepsilon \rangle$ is the mean dissipation rate per unit mass found using the large-scale energy budget rather than the small-scale shear average. Therefore, we define $\nu$ as

$$\nu(t) = \left[ \left( \frac{h}{2.1} \right)^4 \langle \varepsilon \rangle \right]^{1/3}, \tag{38}$$

where, the dissipation rate is determined using the average potential energy $\langle E_p \rangle$ and kinetic energy $\langle K \rangle$ as follows:

$$\langle \varepsilon \rangle = \frac{1}{\langle \rho \rangle L} \frac{d}{dt} \left( \langle \rho \rangle L \left[ \langle E_p \rangle - \langle K \rangle \right] \right). \tag{39}$$

The coefficient 2.1 is a classical value given by Pope[185] to limit the pile-up of energy on small scales. The use of $\eta$ here is justified by the presence of a Kolmogorov cascade in RT-driven flows [186–188]. On 128 CPU cores, it takes 1 hour to obtain 1 simulation, $\sim 65$ hours in total.

**Varied Physical Parameters.** We run simulations with 13 different initializations for five different Atwood number $At \in \{ \frac{3}{4}, \frac{1}{2}, \frac{1}{4}, \frac{1}{8}, \frac{1}{16} \}$. The first set on initial conditions considers varying the mean $\mu$ and standard deviation $\sigma$ of the profile $A(k)$ with $\mu \in \{1, 4, 16\}$ and $\sigma \in \{ \frac{1}{4}, \frac{1}{2}, 1 \}$, the phase (argument of the complex Fourier component) $\phi$ was set randomly in the range $[0, 2\pi)$. The second set of initial conditions considers a fixed mean ($\mu = 16$) and standard deviation ($\sigma = 0.25$) and a varied range of random phases (complex arguments $\phi \in [0, \phi_{max})$) given to each Fourier component. The four cases considered are specified by $\phi_{max} \in \{ \frac{\pi}{128}, \frac{\pi}{8}, \frac{\pi}{2}, \pi \}$.

**Fields present in the data.** Density (scalar field), velocity (vector field).

**References to cite when using these simulations:** [187]

## C.12 `shear_flow`

**Description of the physical phenomenon.** We consider a 2D-periodic incompressible shear flow whose velocity $u = (u_x, u_z)$ (horizontal and vertical) and pressure $p$ are governed by the following Navier-Stokes equation:

$$\frac{\partial u}{\partial t} - \nu \Delta u + \nabla p = -u \cdot \nabla u.$$

where $\Delta = \nabla \cdot \nabla$ is the spatial Laplacian, with the additional constraints $\int p = 0$ (pressure gauge). In order to better visualize the shear, we consider a passive tracer field $s$ governed by the advection-diffusion equation

$$\frac{\partial s}{\partial t} - D \Delta s = -u \cdot \nabla s.$$

We also track the vorticity $\omega = \nabla \times u = \frac{\partial u_z}{\partial x} - \frac{\partial u_x}{\partial z}$ which measures the local spinning motion of the fluid. The shear is created by initializing the velocity $u$ at different layers of fluid moving in opposite horizontal directions.

The fluid equations are parameterized by the Reynolds and Schmidt numbers through the viscosity $\nu$ and the tracer diffusivity $D$

$$\nu = (\text{Reynolds})^{-1},$$
$$D = (\text{Reynolds} \times \text{Schmidt})^{-1}.$$

The Reynolds number is a dimensionless parameter that measures the relative importance of inertial forces to viscous forces. The Schmidt number measures the relative importance of momentum diffusivity and mass diffusivity.

Shear flows are challenging to model and predict due to their inherent instability and the potential for turbulent transition, which is highly sensitive to initial conditions and external perturbations. This instability leads to complex flow phenomena such as Kelvin-Helmholtz instabilities [189], turbulent eddies, and vortex formation, all of which require high-resolution simulations to capture accurately.

**Simulation details.** The data is generated by solving the PDEs through mixed Fourier-Chebychev pseudospectral methods using the Dedalus software [112]. The solution is evolved over time with adaptive time-steps. With 7 nodes of 64 CPU cores, each with 32 tasks running in parallel, it takes $\sim 5$ hours to generate all the data.

**Varied Physical Parameters.** Reynolds $\in \{1e4, 5e4, 1e5, 5e5\}$, Schmidt $\in \{0.1, 0.2, 0.5, 1.0, 2.0, 5.0, 10.0\}$. For initial conditions $n_{\text{shear}} \in \{2, 4\}$ (number of shear), $n_{\text{blobs}} \in \{2, 3, 4, 5\}$ (number of blobs), $w \in \{0.25, 0.5, 1.0, 2.0, 4.0\}$ (width factor of the shear).

**Fields present in the data.** Tracer (scalar field), velocity (vector field), pressure (scalar field).

**References to cite when using these simulations:** [112].

## C.13 `supernova_explosion`

**Description of the physical phenomenon.** The simulations solve an explosion inside a compression of a monatomic ideal gas, which follows the equation of state with the specific heat ratio $\gamma = 5/3$:

$$P = (\gamma - 1)\rho u, \tag{40}$$

where $P$, $\rho$, and $u$ are the pressure, smoothed density, and specific internal energy. The adiabatic compressible gas follows the following equations:

$$\frac{d\rho}{dt} = -\rho \nabla \cdot \boldsymbol{v}, \tag{41}$$

$$\frac{d^2 \boldsymbol{r}}{dt^2} = -\frac{\nabla P}{\rho} + \boldsymbol{a}_{\text{visc}} - \nabla \Phi, \tag{42}$$

$$\frac{du}{dt} = -\frac{P}{\rho} \nabla \cdot \boldsymbol{v} + \frac{\Gamma - \Lambda}{\rho}, \tag{43}$$

where $r$ is the position, $a_\text{visc}$ is the acceleration generated by the viscosity, $\Phi$ is the gravitational potential, $\Gamma$ is the radiative heat influx per unit volume, and $\Lambda$ is the radiative heat outflux per unit volume.

Under a one-dimensional spherical symmetry model [190], an analytic solution describes the propagation of blastwaves in a uniform medium. The time evolution of the radius of the SN shell is written as

$$R(t) = \xi \left( \frac{E}{\rho} \right)^{1/5} t^{2/5}, \tag{44}$$

where $E$, $\rho$, and $\xi$ are the energy injected by SN, the density of the surrounding ISM, and the dimensionless similarity variable, respectively. However, ISM has a large density contrast. Turbulence and cooling form a dense filamentary structure, especially in star-forming regions where SN often occurs. Such structure prevents the blastwave's propagation, and the SN remnants' shells become anisotropic.

**Simulation details.** The simulations are implemented with $N$-body/SPH code, ASURA-FDPS [191, 115, 192] at `https://github.com/FDPS/FDPS`. To solve the hydrodynamic interaction, a DISPH [193] is employed. SPH methods may encounter difficulties resolving contact discontinuities caused by shock waves (such as SN shells) with low mass resolution. Integration timesteps are determined by the resolution and thermal energy [194] so that the blastwave is resolved. Nevertheless, the code has been tested and verified to resolve the shock wave accurately. It can capture the formation of SN shells caused by thermal energy when the mass resolution is finer than 1 solar mass [192, 195]. The gas in simulations has 1 solar metallicity to mimic the environment around the solar system, which causes a strong radiative cooling. For the $128^3$ data, it takes $\sim 3500$ CPU hours on up to 1040 CPU cores to generate all data. For the $64^3$ data, it takes $\sim 3800$ hours on up to 1040 CPU cores to generate all data.

**Varied Physical Parameters.** Initial temperature $T_0$={100K}, Initial number density of hydrogen $\rho_0$ ={44.5/cc}, metallicity (effectively strength of cooling) $Z = \{Z_0\}$.

**Fields present in the data.** Pressure (scalar field), density (scalar field), temperature(scalar field), velocity (vector field).

**References to cite when using these simulations:** [192, 195]

### C.14 `turbulence_gravity_cooling`

**Description of the physical phenomenon.** Similar to `supernova_explosion`, the simulations solve a compression of a monatomic ideal gas, which also follows the equations (40) - (43). To explore different evolutions of ISM under several conditions, simulations are performed with variant initial density, initial temperature, and metallicity with a similar setup to [192]. Metallicity refers to the effectiveness of radiative cooling and heating. In this dataset, richer metallicity mostly has a stronger radiative cooling.

**Simulation details.** Simulations are implemented with $N$-body/SPH code, ASURA-FDPS [191, 115, 192] at `https://github.com/FDPS/FDPS`. A Density-Independent Smoothed Particle Hydrodynamics (DISPH) [193] is employed to solve the hydrodynamic interaction.

The simulations are performed with two resolutions (1 solar mass and 0.1 solar mass) to capture detailed structures in turbulence. First, gas spheres with a total mass of $10^6$ solar mass are generated to make initial gas clouds with turbulence following $\propto v^{-4}$ mimicking star-forming regions. By changing radius, uniform densities are varied in three levels. The initial conditions are constructed using the Astrophysical Multi-purpose Software Environment [196–198]. Radiation is included using the metallicity-dependent cooling and heating functions from 10 to $10^9$ K generated by CLOUDY version 13.5 [199–201]. Assuming the environment of the Milky Way Galaxy, dwarf galaxies, and the early universe, 1 solar metallicity, 0.1 solar metallicity, and 0 metallicity (adiabatic) are adopted. The turbulent spherical clouds are initialized at three different temperatures: 10 K, 100 K, and 1000 K. Details about each simulation time are available on the `README.md` of the dataset.

**Varied Physical Parameters.** Random seeds for generating an initial turbulence velocity field, Initial temperature $T_0$={10K, 100K, 1000K}, Initial number density of hydrogen $\rho_0$ ={44.5/cc, 4.45/cc, 0.445/cc}, metallicity (effectively strength of cooling) $Z = \{Z_0, 0.1Z_0, 0\}$.

**Fields present in the data.** Pressure (scalar field), density (scalar field), temperature (scalar field), velocity (vector field).

**References to cite when using these simulations:** [192].

**C.15** `turbulent_radiative_layer_2D` **and** `turbulent_radiative_layer_3D`

**Description of the physical phenomenon.** The simulations solve the standard fluid equations with an additional energy source term, which removes thermal energy at a rate $t_{\text{cool}}$ which is fastest for intermediate temperatures between the hot and cold phase. The full system of equations solved is given by:

$$\frac{\partial \rho}{\partial t} + \nabla \cdot (\rho \boldsymbol{v}) = 0 \tag{45}$$

$$\frac{\partial \rho \boldsymbol{v}}{\partial t} + \nabla \cdot (\rho \boldsymbol{v} \boldsymbol{v} + P) = 0 \tag{46}$$

$$\frac{\partial E}{\partial t} + \nabla \cdot ((E+P)\boldsymbol{v}) = -\frac{E}{t_{\text{cool}}} \tag{47}$$

$$E = P/(\gamma-1) \quad \text{where} \quad \gamma = 5/3 \tag{48}$$

The major result from these simulations and the corresponding analytic theory is that the total volume integrated radiative cooling is proportional to the net rate of transfer of mass from the hot phase to the cold phase, and that both are proportional to the relative velocity of the phases risen to the 3/4 and the cooling time to the -1/4 power, i.e. $\dot{E}_{\text{cool}} \propto \dot{M} \propto v_{\text{rel}}^{3/4} t_{\text{cool}}^{-1/4}$.

**Simulation details.** 2D data takes 100 CPU core hours on nodes of 48 CPUs to generate all data, while 3D data was generated on 128 core nodes, taking 34560 CPUhours for all simulations.

**Varied Physical Parameters.** $t_{\text{cool}} = \{0.03, 0.06, 0.1, 0.18, 0.32, 0.56, 1.00, 1.78, 3.16\}$.

**Fields present in the data.** Density (scalar field), pressure (scalar field), velocity (vector field).

**References to cite when using these simulations:** [95].

**C.16** `viscoelastic_instability`

**Description of the physical phenomenon.** This dataset contains results from two-dimensional direct numerical simulations between two parallel walls with periodic boundary conditions in the streamwise (horizontal) direction and no velocity at the walls. The governing equations of the problem read,

$$Re(\partial_t \mathbf{u} + \mathbf{u} \cdot \nabla \mathbf{u}) + \nabla p = \beta \Delta \mathbf{u} + (1-\beta)\nabla \cdot \mathbf{T}(\mathbf{C}), \tag{49a}$$

$$\nabla \cdot \mathbf{u} = 0. \tag{49b}$$

We consider FENE-P fluids, where the polymeric stress is related to the conformation tensor $\mathbf{C}$ - an ensemble average of the product of the end-to-end vector of each polymer molecule - via

$$\mathbf{T}(\mathbf{C}) := \frac{1}{Wi}\left(\frac{\mathbf{C}}{1-(\text{tr}(\mathbf{C})-3)/L_{\text{max}}^2} - \mathbf{I}\right). \tag{49c}$$

We consider the evolution equation for the polymer conformation tensor $\mathbf{C}$,

$$\partial_t \mathbf{C} + (\mathbf{u} \cdot \nabla)\mathbf{C} + \mathbf{T}(\mathbf{C}) = \mathbf{C} \cdot \nabla \mathbf{u} + (\nabla \mathbf{u})^T \cdot \mathbf{C} + \varepsilon \Delta \mathbf{C}. \tag{49d}$$

In these equations $\mathbf{u} = (u, v)$ is the velocity with $u$ and $v$ the streamwise and wall-normal velocity respectively, $p$ is the pressure, $\beta := \nu_s/\nu$ is a ratio of kinematic viscosities, where $\nu_s$ and $\nu_p = \nu - \nu_s$ are the solvent and polymer contributions respectively, and $L_{\text{max}}$ is the maximum extensibility of the polymer chains. The half-distance between the plates $h$ and the bulk velocity $U_b$ are used to make the system non-dimensional. The remaining non-dimensional parameters are the Reynolds, $Re := U_b h/\nu$, and Weissenberg, $Wi := \tau U_b/h$, numbers, where $\tau$ is the polymer relaxation time, along with the parameter $\varepsilon := D/U_b h$ which is the dimensionless polymer stress diffusivity.

**Simulation details.** The edge states in the present data set are obtained by bisecting between initial conditions known to reach each attractor. This is done between the laminar state and EIT and between EIT and SAR. The data is generated using the Dedalus codebase [112]. It takes $\sim 1$ day to generate $\sim 50$ snapshots on 32 or 64 CPU cores, 3 months in total.

**Varied Physical Parameters.** Reynold number $Re = 1000$, Weissenberg number $Wi = 50$, $\beta = 0.9$, $\epsilon = 2.10^{-6}$, $L_{max} = 70$.

**Fields present in the data.** pressure (scalar field), velocity (vector field), positive conformation tensor ($c_{xx}^*, c_{yy}^*, c_{xy}^*$ are in tensor fields, $c_{zz}^*$ in scalar fields).

**References to cite when using these simulations:** [96].

# D    Additional Tasks of Interest

The Well contains an enormous diversity of data and can be used for more than forecasting dynamics. We propose a list of additional challenges to be tackled within the Well:

- *Super-resolution:* `MHD` has been downsampled and is available at two resolutions. `supernova_explosion` has been generated at two resolutions. For `MHD` which is downsampled, infer the unresolved scales from the remaining scales. For either, explore generalization from lower resolution training to higher resolution.

- *Transfer across dimensionality:* The same physical phenomenon is represented in 2D and 3D in `turbulent_radiative_layer_2D` and `turbulent_radiative_layer_3D`. Identify approaches for generalizing from cheaper 2D training to more expensive 3D dynamics.

- *Time-steps generalization:* `rayleigh_taylor_instability` simulations for different Atwood numbers have different simulation time-steps. Develop a model trained at a given time-step that can generalize to others.

- *Transfer across a physical parameter range:* Develop a model trained on a restricted range of physical parameters that can generalize to unseen ones which can have different physics behavior.    Datasets: `active_matter`, `gray_scott_reaction_diffusion`, `rayleigh_benard`, `viscoelastic_instability`, `shear_flow`, `euler_multi_quadrants`  generate data across ranges of parameters that can easily be filtered in the provided dataset object.

- *Steady-state  prediction:* `convective_envelope_rsg` and `gray_scott_reaction_diffusion` eventually reach a steady-state. Predict this steady-state from initial conditions.

- *Stable long-term forecasting:* Each trajectory of `planetswe` is rolled out for three model years. Develop models that can produce stable predictions in the sense that the forecasted states follow the same distribution as the simulated system at long time horizons.

- *Sensitivity to initial conditions:* `rayleigh_benard` Simulations form convective cells at certain positions within the domain over time. These positions are highly sensitive to small variations in the initial conditions.

- *Inverse-scattering problem:* `acoustic_scattering` and `helmholtz_staircase` contain forward simulations of acoustic waves scattering in response to different material densities. Try instead predicting the material densities from the evolution of the pressure fields.

- *Simulation acceleration:* `post_neutron_star_merger` and `turbulence_gravity_cooling` are enormously expensive simulations taking months to generate. Accurate predictions here can constitute an enormous speed-up relative to the generating process.

# E    Benchmarking Details

## E.1    Standard Methodology

The preliminary benchmarks included in the Well are intended to demonstrate the value of new, more challenging tasks for pushing the field forward. As the focus of this work is on the data, our benchmarking methodology is designed to be representative of a generic standard practice in the field both in terms of design choices and computational resources. With that in mind, all benchmarks were performed with the following procedure:

- Baseline models were scaled to approximately 15-20 million parameters.

- Batch size was chosen to maximize GPU memory consumption for a given dataset.

- AdamW was used for all experiments with the PyTorch default WD of .01. We performed a coarse learning rate search over $\{1 \times 10^{-4}, 5 \times 10^{-4}, 1 \times 10^{-3}, 5 \times 10^{-3}, 1 \times 10^{-2}\}$. The run with the best validation VRMSE was used for subsequent reporting (see Table 6) and evaluated on the test set (see Table 2.

- All models and datasets were trained using Mean Squared Error averaged over fields and space during training.

- Boundary conditions were handled naively according to model architecture. Fourier domain convolutions implicitly used periodic boundaries while spatial domain convolutions utilized standard zero padding.

- All runs were time-limited to 12 hours on a single Nvidia H100 GPU. Due to the size of these datasets, this intentionally gave an advantage to faster models. As such, we used recent, optimized libraries wherever possible and avoided cutting-edge architectures without optimized GPU kernels.

- Single precision was used for all experiments as several datasets encountered stability issues with mixed or low precision training.

## E.2 Models

We opted to stick with time-tested models that are widely used in applications and that natively extend to 3D. This is not intended to be an exhaustive baseline, but rather provide a starting point for the community to use in their own studies. The Fourier Neural Operator [97, FNO] and U-net [99] are among the most widely used models for data driven surrogates. While neither can fairly be called state of the art at this point, they have demonstrated robustness across many problems and are common starting points for practitioners. The TFNO [202] is a more recent tensor-factorized variant of the FNO that improves scalability. We additionally felt that the 2015 variant of the U-net with MaxPool layers and Tanh activations was lacking many recent improvements and so replaced the convolutional blocks with a modern ConvNext [100] architecture for fairer evaluation.

As mentioned in the previous section, all models were scaled to obtain approximately 15-20 million parameters for 2D models. We prioritized reaching this with adjustments to depth or width rather than filter size or downsampling rates. The hyperparameter settings that allowed us to reach these are as follows:

- FNO
  - Spectral filter size (modes) - 16
  - Hidden dimension - 128
  - Blocks - 4
- TFNO
  - Spectral filter size (modes) - 16
  - Hidden dimension - 128
  - Blocks - 4
- U-net Classic
  - Spatial filter size - 3
  - Initial dimension - 48
  - Blocks per stage - 1
  - Up/Down blocks - 4
  - Bottleneck blocks - 1
- CNextU-net
  - Spatial filter size - 7
  - Initial dimension - 42
  - Blocks per stage - 2
  - Up/Down blocks - 4
  - Bottleneck blocks - 1

## E.3 Metrics

We evaluate the performance of our models using a diverse set of spatial metrics, namely:

- The mean squared error (MSE): for two spatial fields $u$ and $v$ it is defined as:

$$\mathrm{MSE}(u,v) = \langle |u-v|^2 \rangle,$$

where $\langle \cdot \rangle$ denotes the spatial mean operator. We also consider its variant the root mean squared error (RMSE) that is the square root of the MSE.

- The normalized mean squared error (NMSE): it corresponds to the MSE normalized by the mean square value of the truth, that is:

$$\text{NMSE}(u,v) = \langle |u-v|^2 \rangle / (\langle |u|^2 \rangle + \epsilon),$$

where $\epsilon = 10^{-7}$. The term $\epsilon$ prevents division by zero in cases where $\langle |u|^2 \rangle$ reaches zero. We also consider its square root variant called the NRMSE.

- The variance scaled mean squared error (VMSE): it is the MSE normalized by the variance of the truth

$$\text{VMSE}(u,v) = \langle |u-v|^2 \rangle / (\langle |u-\bar{u}|^2 \rangle + \epsilon).$$

We chose to report its square root variant, the VRMSE:

$$\text{VRMSE}(u,v) = \left( \langle |u-v|^2 \rangle / (\langle |u-\bar{u}|^2 \rangle + \epsilon) \right)^{1/2}.$$

Note that, since $\text{VRMSE}(u,\bar{u}) \approx 1$, having $\text{VRMSE} > 1$ indicates worse results than an accurate estimation of the spatial mean $\bar{u}$.

- The maximum error ($L^\infty$):

$$L^\infty(u,v) = \max |u-v|$$

.

- The binned spectral mean squared error (BSMSE): it is the MSE after bandpass filtering of the input fields on a given frequency band $\mathcal{B}$, that is:

$$\text{BSMSE}_\mathcal{B}(u,v) = \langle |u_\mathcal{B} - v_\mathcal{B}|^2 \rangle,$$

where $u_\mathcal{B} = \mathcal{F}^{-1}[\mathcal{F}[u]\mathbf{1}_\mathcal{B}]$, with $\mathcal{F}$ the discrete Fourier Transform and $\mathbf{1}_\mathcal{B}$ the indicator function over the set of frequencies $\mathcal{B}$. For each dataset, we define three disjoint frequency bands $\mathcal{B}_1$, $\mathcal{B}_2$, and $\mathcal{B}_3$ corresponding to low, intermediate, and high spatial frequencies, respectively. In practice, these bands are defined by partitioning the frequencies based on the magnitudes of their wavenumbers, which are split evenly on a logarithmic scale.

- The binned spectral normalized mean square error is a variant of the previous metric normalized to bin energy of the target:

$$\text{BSNMSE}_\mathcal{B}(u,v) = \langle |u_\mathcal{B} - v_\mathcal{B}|^2 \rangle / \langle |v_\mathcal{B}|^2 \rangle,$$

thus a value of 1 or more indicates that the model would have performed better if it had predicted coefficients of zero corresponding to that scale. This is used in Figure 6 for instance to make the rollout quality more immediately visually interpretable.

### E.4 Results

We report the one-step VRMSE on the test sets in Table 2 as well as the time-averaged losses by window in Table 3, for the models performing best on the validation set in Table 5. In several cases, the simple, generic training approach works quite poorly. We choose VRMSE as the reporting metric as it has the clear interpretation that scores above 1.0 indicates one could have improved the result by predicting the non-spatially varying mean of the target. This is not the same as predicting the population mean, but it is a significantly easier task that predicting the spatially varying target.

When we dig deeper into individual datasets as we do in Figure 6, we can see that performance is not uniform across fields. Even when overall performance is poor, individual fields may obtain good accuracy. Perhaps this is in part due to the use of unnormalized losses during training which could support the use of normalized losses for general surrogate modeling tasks.

Interestingly, though also predictably, we see the model is better able to track the evolution of low frequency modes over time while high frequency modes diverge relatively quickly. The metrics included in the Well pipeline provide valuable insights like this into training and developing new architectures.

More generally, certain datasets proved particularly challenging due to either computational limitations or inherent complexities in their dynamics. For the following datasets, the training could only be done on less than 5 epochs within 12 hours (see Table 6): `convective_envelope_rsg` (544GB), `euler_multi_quadrants` (4.9TB), `turbulence_gravity_cooling` (793GB), `turbulent_radiative_layer_3D` (711GB). Non-time limited training could improve the results.

| Dataset | FNO | TFNO | U-net | CNextU-net |
|---|---|---|---|---|
| acoustic_scattering (maze) | 0.5033 | 0.5034 | 0.0395 | **0.0196** |
| active_matter | 0.3157 | 0.3342 | 0.2609 | **0.0953** |
| convective_envelope_rsg | 0.0224 | **0.0195** | 0.0701 | 0.0663 |
| euler_multi_quadrants (periodic b.c.) | 0.3993 | 0.4110 | 0.2046 | **0.1228** |
| gray_scott_reaction_diffusion | 0.2044 | **0.1784** | 0.5870 | 0.3596 |
| helmholtz_staircase | 0.00160 | **0.00031** | 0.01655 | 0.00146 |
| MHD_64 | 0.3352 | 0.3347 | 0.1988 | **0.1487** |
| planetswe | **0.0855** | 0.1061 | 0.3498 | 0.3268 |
| post_neutron_star_merger | 0.4144 | **0.4064** | – | – |
| rayleigh_benard | 0.6049 | 0.8568 | 0.8448 | **0.4807** |
| rayleigh_taylor_instability (At = 0.25) | 0.4013 | **0.2251** | 0.6140 | 0.3771 |
| shear_flow | 0.4450 | **0.3626** | 0.836 | 0.3972 |
| supernova_explosion_64 | 0.3804 | 0.3645 | 0.3242 | **0.2801** |
| turbulence_gravity_cooling | 0.2381 | 0.2789 | 0.3152 | **0.2093** |
| turbulent_radiative_layer_2D | 0.4906 | 0.4938 | 0.2394 | **0.1247** |
| turbulent_radiative_layer_3D | 0.5199 | 0.5174 | 0.3635 | **0.3562** |
| viscoelastic_instability | 0.7195 | 0.7021 | 0.3147 | **0.1966** |

Table 5: Dataset and model comparison in VRMSE metric on the validation sets, best result in **bold**. VRMSE is scaled such that predicting the mean value of the target field results in score of 1.

| Dataset | FNO | TFNO | U-net | CNextU-net |
|---|---|---|---|---|
| acoustic_scattering (maze) | 1E-3 (27) | 1E-3 (27) | 1E-2 (26) | 1E-3 (10) |
| active_matter | 5E-3 (239) | 1E-3 (243) | 5E-3 (239) | 5E-3 (156) |
| convective_envelope_rsg | 1E-4 (14) | 1E-3 (13) | 5E-4 (19) | 1E-4 (5) |
| euler_multi_quadrants (periodic b.c.) | 5E-4 (4) | 5E-4 (4) | 1E-3 (4) | 5E-3 (1) |
| gray_scott_reaction_diffusion | 1E-3 (46) | 5E-3 (45) | 1E-2 (44) | 1E-4 (15) |
| helmholtz_staircase | 5E-4 (132) | 5E-4 (131) | 1E-3 (120) | 5E-4 (47) |
| MHD_64 | 5E-3 (170) | 1E-3 (155) | 5E-4 (165) | 5E-3 (59) |
| planetswe | 5E-4 (49) | 5E-4 (49) | 1E-2 (49) | 1E-2 (18) |
| post_neutron_star_merger | 5E-4 (104) | 5E-4 (99) | - | - |
| rayleigh_benard | 1E-4 (32) | 1E-4 (31) | 1E-4 (29) | 5E-4 (12) |
| rayleigh_taylor_instability (At = 0.25) | 5E-3 (177) | 1E-4 (175) | 5E-4 (193) | 5E-3 (56) |
| shear_flow | 1E-3 (24) | 1E-3 (24) | 5E-4 (29) | 5E-4 (9) |
| supernova_explosion_64 | 1E-4 (40) | 1E-4 (35) | 5E-4 (46) | 5E-4 (13) |
| turbulence_gravity_cooling | 1E-4 (13) | 5E-4 (10) | 1E-3 (14) | 1E-3 (3) |
| turbulent_radiative_layer_2D | 5E-3 (500) | 1E-3 (500) | 5E-3 (500) | 5E-3 (495) |
| turbulent_radiative_layer_3D | 1E-3 (12) | 5E-4 (12) | 5E-4 (13) | 5E-3 (3) |
| viscoelastic_instability | 5E-3 (205) | 5E-3 (199) | 5E-4 (198) | 5E-4 (114) |

Table 6: Optimal learning rate and number of training epochs (in parenthesis) to obtain the VRMSE validation loss reported in Table 5.

