# OpenReview forum: "The Well: a Large-Scale Collection of Diverse Physics Simulations for Machine Learning"
_NeurIPS.cc/2024/Datasets_and_Benchmarks_Track — NeurIPS 2024 Track Datasets and Benchmarks Poster_

### Official Review · Reviewer_FPen · 2024-07-25
**Review of the Well, a dataset of PDE solutions**

**Rating:** 6
**Confidence:** 3

**Review:**

This work distinguishes itself from the many PDE solution datasets available due to its volume of data, variety of methods and degree of specialization/connection to dynamical systems of interest. Whereas many existing datasets use rather pedagogical PDEs, this work compiles specialized dynamical systems, which are of interest to: 1. Peers working in the same specialized fields. 2. Generalists looking to test methods past the canonical PDEs included in every other dataset. There are a few things missing, such as consideration of running time and in distribution/out of distribution evaluation as metrics for learning methods. In general, this work exposes a lot of high quality data in many specialized fields for general practitioners to evaluate their model performance.

**Strengths:**

* Long list of time varying dynamical systems examined, each varying drastically between one another.
* Since each set of PDE solutions was compiled by domain experts, there is a high level of confidence in the data generation (ground truth simulation) methods.
* Large volume of data.

**Additional Feedback:**

N/A

**Clarity:**

The paper is very well written and the supplemental clearly outlines technical details for each dynamical system.

**Correctness:**

The dataset is constructed in a sound way, there are enough samples for each dynamical system (and large overall volume of data), and each set of ground truth data is generated by a domain expert.

**Documentation:**

The supplemental material includes all technical details necessary for users of this dataset, and the breakdown of the dataset is clearly done in the main text (number of trajectories, domain dimensions, time steps, etc).

**Limitations:**

* The main limitation (limitation of domain size/shape) has been considered and with such a dataset you always have to make tradeoffs, but I think that all of the ones done here are very sensible.
* Another limitation is a lack of consideration for running time as another metric for benchmarking methods, particularly compared to traditional numerical solvers.
* The paper mentions that the train/test split is random, so presumably they come from a similar distribution. A set of solutions out of distribution would provide for a more compelling test, although this would vary for each PDE.

**Opportunities For Improvement:**

* This dataset does not give any indication of performance and runtime of traditional methods. The supplementary material gives the times used to generate the entire dataset, but there is no reference point given for a more traditional numerical solver with, perhaps, acceptable accuracy but faster runtime than the ground truth generation.
* This is outside of the scope of this work, but the dynamical systems are all on simple geometries.

**Relation To Prior Work:**

There has been some discussion of prior work, though not comprehensive with respect to what each of them try to achieve. Many PDE benchmarks task themselves with better evaluation metrics, dataset split distribution and runtime metrics that are not considered in this work and these are not mentioned in the context of prior contributions.

**Summary And Contributions:**

The Well is a collection of numerical solutions to dynamical systems used to benchmark neural network methods. They are all solved on a regular grid and are time varying, with the time steps coarsened to keep the data size manageable. Even so, the total size of the dataset is 15TB, so it represents a wealth of data to evaluate machine learning methods for PDEs.

---

> ### Author Rebuttal · Authors · 2024-08-15
>
> We'd first like to thank the reviewer for their time and effort. Your feedback is valuable and in this rebuttal, we hope to address your concerns and open up discussion into how we can use your feedback to improve the paper. If we are able to address the reviewer's concerns, we ask that the reviewer adjust their scores accordingly.
>
> Overall, it seemed that the reviewer had three major concerns, so we divide our response into these three areas:
>
> > This dataset does not give any indication of performance and runtime of traditional methods. The supplementary material gives the times used to generate the entire dataset, but there is no reference point given for a more traditional numerical solver with, perhaps, acceptable accuracy but faster runtime than the ground truth generation.
>
> This is a point we feel merits deeper discussion. As the reviewer mentioned, this data is available in the paper, but it is aggregated to the full dataset level and not highlighted as a basis for comparison. It would be straightforward to add this information to Table 1 so users of the dataset can use them in comparisons. However, this exclusion was an intentional decision on our part. We are willing to adjust this if the reviewers feel strongly, but first we would like to describe our reasoning.
>
> In short, we did not feel this comparison would be fair and believed that emphasizing the comparison would lead to specious claims of outperforming numerical methods. The two main issues are:
>  - Hardware imbalance - As mentioned in the appendix, our results are from a mix of a clusters and simulation codes with some CPU and some GPU based. From a cost perspective, it made the most sense to run many of the CPU-based simulations on older hardware. The most common CPU used was an Intel Icelake Xeon Platinum CPU, a 2019 CPU. On the other hand, our learned models were typically trained on NVIDIA H100 GPUs, a 2022 model. At current prices, the H100 has similar cost to about 20 Icelakes. Fair comparisons on run-time would require alignment both in terms of hardware and also in terms of energy expenses.
> - Optimization vs generalization - Building a dataset requires different implementation choices compared to production use. For example, several datasets were produced using software that employs adaptive mesh refinement (AMR). Interpolating these adaptive meshes back onto a uniform grid was quite slow in practice since it is an atypical workflow for scientists in those areas and therefore poorly optimized. We found it to be faster to simply execute the simulation at maximum resolution on larger nodes. This is not a fair comparison in either direction as the numerical method is hamstrung by an unusual requirement while the learned model is unable to benefit from AMR.
>
> Ultimately, our focus in the Well was providing a large volume of interesting, diverse, and challenging dynamics data. The Well is a large collaboration between a large number of scientists and computing centers and while that diversity has allowed us to build what we feel is a very valuable dataset, it also restricts the level of uniformity in compute details. We feel truly fair time-comparisons with numerical methods require a more limited scope in order for the level of control required to be feasible. When possible, we do provide the generation code for the data so that users could potentially explore this type of analysis themselves.
>
> > The paper mentions that the train/test split is random, so presumably they come from a similar distribution. A set of solutions out of distribution would provide for a more compelling test, although this would vary for each PDE.
>
> The provided baselines were designed to give users a sense of how “vanilla” training recipes perform on the Well. Due to the limited space format, we kept the baselines uniform in the sense that a single workflow could apply to all of the data. In practice, our assumption is that most usage of the Well will be specialized - users will take the datasets that allow them to best illustrate their point rather than using the full data for every paper.
>
> However, out-of-distribution testing is explicitly supported via config files or command line overrides. If we look at the Dataset class in our codebase (https://anonymous.4open.science/r/the_well-4DB3/the_well/benchmark/data/datasets.py - lines 173,174), we can see lists of filters that allow users to filter datasets based on their names. The naming schema of files in the Well includes the generating parameters specifically to enable these types of flexible train/eval rules.
>
> For instance, one file in pattern formation is named - *pattern_formation_bubbles_F_0.098_k_0.057.hdf5*. One can therefore choose to make a training set consisting of “bubbles” or feed rate of .098 or kill rate of .057 and explicitly choosing an out-of-distribution validation set. This is fully controllable from the configuration files, so it is very easy for users to construct this type of scenario.
>
> Based on your feedback, we plan to add an example configuration file and documentation to demonstrate how one would do this.
>
> > There has been some discussion of prior work, though not comprehensive with respect to what each of them try to achieve. Many PDE benchmarks task themselves with better evaluation metrics, dataset split distribution and runtime metrics that are not considered in this work and these are not mentioned in the context of prior contributions.
>
> The related work was certainly a section that suffered due to page limitations. We’d be happy to expand on this with an additional page available if accepted and would be interested in any particular recommendations the reviewer might have.
>
>
> ___
>
> Once again, we greatly appreciate the reviewer's feedback! This discussion has been extremely useful for us and we hope that we were able to address the reviewer's concerns. Please let us know if you have further concerns or comments on the discussion points raised.

---

> > ### Comment · Reviewer_FPen · 2024-09-01
> > **Comment**
> >
> > I am satisfied with the responses, I think the choices made are reasonable. I'll increase the score to 7.
> > As for additional literature for prior work, there have been a few papers in prior years at NeurIPS Datasets and Benchmarks Track that you could add.

---

### Official Review · Reviewer_XnXV · 2024-07-25
**Review of "The Well: a Large-Scale Collection of Diverse Physics Simulations for Machine Learning"**

**Rating:** 7
**Confidence:** 4
**Correctness:** Yes, the claims made in the submissio…
**Clarity:** The paper is well written and communi…

**Review:**

This manuscript is well written with the supplementary information also being a valuable resource to document all of the intricacies of how data for each problem was created. The dataset is structured well with enough metadata and a benchmarking library to make building ML based surrogate models for scientific applications approachable for ML researchers that are not necessarily domain experts. The paper also presents some baselines and metrics but those are mainly meant to showcase what is possible with this dataset and are not meant to be exhaustive evaluation.

One of the main criticisms of the dataset is that some of the problems are motivated by real applications but not generated in a manner that is useful for the end applications. For example, the shear flows and the rayleigh benard problems are very useful turbulent flow problems with many applications but the data is generated in a 2D context when all of the real world applications require a 3D context for accurate representation of the turbulent flow dynamics like vortex stretching, etc. Similarly, the plots of the turbulent_radiative_layer_2D data seem to suggest that the periodic boundary conditions used in this problem cause some mode locking in the spanwise direction artificially constraining the dynamics of the problem.

Some other comments:
- line 13: poses -> posed
- line 73: No link to the benchmarking library is provided
- rayleigh_benard: Why 2D when real convective phenomena and the turbulent structures involved are always 3D?
- shear_flow: Again, why 2D when shear flows in reality always have 3D turbulent structures after the initial linear growth phase?
- Section 3.1.6: How do you define an "ideal" low-pass filter? Spectral filtering?
- 3.1.7 pattern_formation: This seems like too generic of a name. Something more specific would be nicer so it can be referred to in isolation.
- Figure 5, turbulent_radiative_layer_2D: Is the spanwise domain size here sufficient? From the plots, it looks like the spanwise length scales are confined to the domain size and that the periodic boundary conditions are influencing the behavior quite significantly.
- Section 4, "time-boxed to 12 hours on a single NVIDIA H100": Although this is a practical choice to limit compute time, it would be good to also comment on how hyperparameters for each model were chosen. For example, one model could be chosen to have a large number of parameters and so each iteration time is large and 12 hours is not sufficient to learn effectively causing the skill of that model to be particularly low?
- line 185, VRMSE: Is this the appropriate metric for all of the problems? It would seem that for some of the problems with chaotic dynamics, some more statistical metrics rather than a pointwise error would be more meaningful?
- line 216, super-resolution: How can this be done without having access to the original high resolution fields?

**Strengths:**

The main strengths of this manuscript is the new large dataset that encompasses many domains of scientific computing to create a larger scale benchmark dataset than has previously been available. The details of how the data is formatted and stored, how metadata and other supplementary information is made available with the data itself and the availability of a benchmarking library with unified training interfaces is a great enabler for any researchers looking to tackle these problems.

**Additional Feedback:**

Nothing beyond the feedback above.

**Documentation:**

The dataset or benchmarking library links are not provided, so as it stands, there is no documentation.

**Ethics:**

No specific concerns.

**Limitations:**

One of the main criticisms of the dataset is that some of the problems are motivated by real applications but not generated in a manner that is useful for the end applications. For example, the shear flows and the rayleigh benard problems are very useful turbulent flow problems with many applications but the data is generated in a 2D context when all of the real world applications require a 3D context for accurate representation of the turbulent flow dynamics like vortex stretching, etc. Similarly, the plots of the turbulent_radiative_layer_2D data seem to suggest that the periodic boundary conditions used in this problem cause some mode locking in the spanwise direction artificially constraining the dynamics of the problem.

These limitations should ideally be addressed by updating the dataset or by explicitly calling these out in the manuscript, especially since one of the stated goals of the paper is to address the gap currently in existing datasets in terms of the complexity and applicability to real world use cases.

**Opportunities For Improvement:**

In terms of the metrics used for evaluation of the baseline models, only a variance scale RMSE is reported. Is this the appropriate metric for all of the problems? It would seem that for some of the problems with chaotic dynamics, some more statistical metrics rather than a pointwise error would be more meaningful?

**Relation To Prior Work:**

A section on prior work is included in the paper and there is sufficient discussion about how this works differs from previous work.

**Summary And Contributions:**

This manuscript introduces a large dataset containing simulation data from 16 different problems in scientific applications totaling to 15TB of data. The datasets are designed well in a way that is easy to consume in ML pipelines. The paper also claims to provide a benchmarking library with PyTorch based utilities including a unified training pipeline. However, no link or pointer to this benchmarking library is provided. Many of the problems in this dataset are designed with domain specific usecases in mind. The data is structured as uniformly sampled N-D arrays which makes it convenient to try many network architectures easily. In addition, most problems also sample the parameter space and temporal dynamics sufficiently to provide enough sample diversity that is necessary for ML methods. Some initial baselines using common neural operator and convolutional architectures are presented including evaluation of those methods based on a defined VRMSE metric. The analysis also sheds some light into the spectral characteristics of the error growth in time which is insightful and can be used to inspire future work.

---

> ### Author Rebuttal · Authors · 2024-08-15
>
> First off we’d like to thank the reviewer for their time and effort. The reviewer's feedback is invaluable in helping us improve the paper. We hope that in this rebuttal we can address several of the concerns raised to help the reviewer feel confident in their evaluation.
>
> We’ll divide our response between several larger points that require more description and smaller points that can addressed as bullets.
>
> > Documentation
>
> We apologize for the confusion here. The repository was actually provided in a link deep in Appendix B2 (“Data Access”). In the full release, the link will be included in a footnote on the first page. Since the link has expired, we’ve regenerated it here: https://anonymous.4open.science/r/the_well-4DB3/README.md . The github contains both download scripts and the benchmarking library.
>
> > Realism of 2D problems.
>
> This is a good point and we’ll update the limitations and wording to be clearer. In designing this benchmark, we made an effort to balance accessibility and realism. As the reviewer points out, real-world turbulence problems are fundamentally 3D and there are known differences between simulated 2D and real-world 3D turbulence. However, 2D still adds value. Processing 2D data is much cheaper than 3D data and it’s unlikely that a method which works poorly on 2D data will improve on 3D. Having good 2D benchmarks can save a lot of time in the development of full 3D methods.
>
> > Similarly, the plots of the turbulent_radiative_layer_2D data seem to suggest that the periodic boundary conditions used in this problem cause some mode locking
>
> This is a very observant point raised by the reviewers, but we are confident in our setup for several reasons. From a physics perspective this setup is meant to capture the interface between a cold dense clump of gas and a hotter dilute medium moving relative to it. This full problem spans a huge range of spatial and temporal scales that are infeasible to capture in a single simulation, which is why we have just focused on the interface itself.  The clouds we are interested in studying with these simulations are finite in size, which can be replicated by having a finite spanwise domain size comparable to the size of the cold cloud itself and using periodic boundaries which together limit the largest eddies or unstable modes. It has been demonstrated in previous works [1,2] that the crucial dimensionless number is (L / (velocity*tcool)) where L is the spanwise domain size, velocity is the velocity difference between hot and cold gas, and tcool is the characteristic timescale for cooling. Using a larger or smaller box is exactly equivalent to using a smaller or larger velocity or tcool (up to the limit when the velocity becomes supersonic).
>
> Furthermore, regardless of the first argument, from a pure test of numerical methods perspective, this setup is well posed and presents a useful and distinct test of the interplay of various physical mechanisms.
>
> [1] https://ui.adsabs.harvard.edu/abs/2020ApJ...894L..24F/abstract
>
> [2] https://ui.adsabs.harvard.edu/abs/2021MNRAS.502.3179T/abstract
>
> >  line 185, VRMSE: Is this the appropriate metric for all of the problems?
>
> In general, no, but the reporting choices are a compromise based on space limitations. Our emphasis in this submission was on the data and building flexible tools to interact with the data. The particular numerical results are primarily to give a sense of how difficult the problems are for “vanilla” baselines - generic tools applied off the shelf - which we hope motivates the development of more powerful or more specialized new tools.
>
> However, as we show in Figure 6, the (now) provided code is evaluating and logging more than just VRMSE.  Outside of potential foundation model usage, we foresee users picking a few datasets that emphasize the problems their work is trying to address rather than utilizing the full Well. In these cases, the Well’s benchmarking code is set up to facilitate the more interesting types of analysis we see in Figure 6. For public release, we plan on adding additional demonstrations of how to add or adjust the metrics computed in a training run.
>
> >  Section 4, "time-boxed to 12 hours on a single NVIDIA H100"...
>
> This is another good point. While our procedures are described in E.1, it would be good to highlight these caveats in the main text. Overall, as we describe in E.1, our goal in the benchmarking was to employ conventional “naive” strategies to illustrate where out-of-the-box performance lands in the Well. As such, we picked sizes and tasks based on what we observed in the literature and adapted models to those conventions. Within that size range, the time-boxing approach intentionally gives advantages to faster models, but it is also likely true that more tuning around model size might have found better cost/performance trade-offs.
>
> Smaller comments
> - “line 13: poses -> posed” - Thank you. Fixed.
> - “line 73 - benchmarking” - It looks like the anonymized link expired, so we’ve refreshed it - https://anonymous.4open.science/r/the_well-4DB3/README.md. Since this is single-blind reviewing, the anonymous link is just for access before we make the real Github public. All benchmarking tools can be found under the_well/benchmark.
> - “Ideal low pass filtering” - Yes, this is ideal in the spectral sense. The low pass filtering is a brick wall filter/spectral truncation. If the reviewer feels this usage is atypical, we’re happy to rephrase this as spectral truncation.
> - “3.1.7 pattern_formation:” - Good point. We will rename this Gray-Scott Pattern Formation to be more specific.
> - line 216, super-resolution:  - The MHD dataset under discussion here is provided in 256^3 and 64^3 resolution where the 64^3 is downsampled from the 256^3.
>
> Again, we greatly appreciate the reviewer's suggestions. We hope that we were able to clear up some of the reviewer's concerns. If any concerns remain, please raise them and we will be happy to discuss further.

---

### Official Review · Reviewer_b6xs · 2024-08-17

**Rating:** 7
**Confidence:** 4
**Correctness:** I don't have any correctness concerns…
**Clarity:** Put simply, yes. This is one of the s…

**Review:**

At a high level, I find that from a "datasets paper" perspective the submission does a great job of introducing the problem, justifying its importance, and providing a clear and complete description of the datasets and their construction (especially in its supplementary material). From a "benchmark paper" perspective, the submission remains at a somewhat superficial level, but since the paper appears to be framed mostly as a dataset paper, this does not detract too much from its strengths.

**Strengths:**

* The abstract and introduction are crystal clear. They do an excellent job of conveying high-level details on the surrogate modelling problem and justifying its importance and difficulty.
* The application itself is very compelling and of practical importance for a wide variety of scientific domains.
* The dataset visualizations (Figures 1 through 5) are a nice touch: they help get an intuitive feel for the 16 different simulated physical systems.
* The description of each dataset is clear and concise, and the supplementary material provides comprehensive details on each dataset.

**Additional Feedback:**

I believe there is a typo in the abstract: should "new challenges poses" read "new challenges pose**d**"?

**Documentation:**

The dataset collection is appropriately documented. The supplementary material goes to great lengths to describe each simulated numerical system in detail using equations and pseudocode and providing all relevant physical parameters.

**Ethics:**

I don't have any ethical concerns to share.

**Limitations:**

The submission does a good job of discussing the limitations of its specific choice of datasets.

**Opportunities For Improvement:**

* I was able to infer from contextual clues, but it would be helpful to spell out explicitly what the surrogate model is tasked with doing, i.e., predict the future state of a simulated physical system given a history of previous states.
* The paper could use clarification on what design choices are prescribed for standardized evaluation and what design choices are left up to dataset users. For instance, the main text mentions that the baselines are trained to predict the next snapshot of a given evaluation from a "short history" but does not elaborate on how that translates into a specific number of previous time steps or whether controlling for the "context" length is necessary in order to make a fair comparison across approaches. Similarly, the time limit is set to 12 hours but little is said on the rationale behind this number and whether the 12 hour limit is a prescription for all users of the dataset to compare on an equal footing.
* The choice of VRMSE as an evaluation metric is not discussed. Is it a standard metric used in surrogate model evaluation? If so, a simple sentence mentioning that and providing a few citations should suffice. If not, the authors should elaborate on why it is an appropriate evaluation metric.

**Relation To Prior Work:**

I am not familiar with related work in the area, so I can't comment on the comprehensiveness of the cited works, but the submission clearly describes its relationship to the related works it cites.

**Summary And Contributions:**

The submission introduces a collection of datasets ("the Well") of numerical simulations of spatiotemporal physical systems with the aim of driving forward progress in the development of surrogate models. It consists of 16 distinct datasets totalling 15TB of data for domains such as biological systems, fluid dynamics, acoustic scattering, magneto-hydrodynamic simulations, and more. The datasets are canonically partitioned into training, validation, and test splits.

The submission also provides a PyTorch interface for the training and evaluation of surrogate models as well as a set of baselines (Fourier Neural Operator, Tucker-Factorized FNO, U-net, and CNextU-net) that are evaluated in terms of Variance Scaled Root Mean Squared Error (VRMSE).

The authors report that although CNextU-Net outperforms the other baselines on 11 out of 16 experiments, all naive time-boxed surrogate models perform poorly on the benchmark tasks, and that in many cases the surrogate models fail to outperform a mean prediction after the budgeted 12 hours.

---

> ### Author Rebuttal · Authors · 2024-08-20
>
> We’d like to thank the reviewer for their feedback. It has been immensely valuable and we aim to use it to improve the paper. In this rebuttal, we will address several of the main points and hope we can clarify some points of confusion and address your remaining concerns.
>
> >From a "benchmark paper" perspective, the submission remains at a somewhat superficial level, but since the paper appears to be framed mostly as a dataset paper, this does not detract too much from its strengths.
>
> We agree with this assessment. Our submission is primarily focused on introducing a new dataset. The benchmarking is intended to demonstrate where taking a fully “off-the-shelf” approach to the problem would land rather than an extensive evaluation of current methods. We believe this dataset will enable more robust benchmarking going forward and we include our benchmarking code which interacts with multiple models and sub-datasets to facilitate this.
>
> > I was able to infer from contextual clues, but it would be helpful to spell out explicitly what the surrogate model is tasked with doing
>
> Thanks for the feedback. If accepted, we will use part of the additional page in the camera ready version to add a paragraph to the beginning of Section 3 describing surrogate models in more conventional ML language, tying the idea of data-driven emulators with the ML task of predicting future states (either autoregressively or in bulk) from current or historical states.
>
> > The paper could use clarification on what design choices are prescribed for standardized evaluation and what design choices are left up to dataset users…
>
> This is a good point. We’ll clarify our choices further. In the original submission, we included this in Appendix E.1, but it’s important enough that it should be stated in the main body as well (likely in brief with a reference to the appendix for more detail). Our goal in including benchmarks was two-fold:
>
> 1. We wanted to show where off-the-shelf approaches land today - we assume a moderate compute budget with architectures that are proven in the space but not specialized to our data using a standard regression loss for training. While there are a few cases where this is sufficient for strong performance, our results indicate that in general, the Well is far from loss saturated here which is very exciting - it shows that there is a lot of progress to be made before these problems become trivial.
>
> 2. We wanted to demonstrate how to use our benchmarking tools. The code and config files for all benchmarks we’ve run are included and accessible to readers. This shows how to use a variety of models and datasets. While we report 1-step VRMSE in the table primarily due to space limitations, in Figure 6 we also show an example of more interesting types of analysis that can be performed using other metrics tracked in our training pipelines.
>
> While we are not trying to provide a strict framework for how users should evaluate results on the Well, we agree that it would be useful to also include a “benchmarking considerations” section to explain issues like history length or training time. We’ll add this discussion in the supplementary materials.
>
> > The choice of VRMSE as an evaluation metric is not discussed. Is it a standard metric used in surrogate model evaluation?
>
> Good point. This is actually a choice on which we slightly deviate from established literature where nRMSE is significantly more common. We’ll briefly explain our reasoning here with an example and add a short appendix section describing this along with a reference in the main text.
>
> nRMSE, the square root of $nMSE(y, \hat y) = \frac{{\mathbb E[(y - \hat y)^2]}}{{\mathbb E[y^2]} + \epsilon}$, is currently one of the more common deterministic metrics used for short-term predictions. The reason is not due to the physics themselves, but rather that many physical fields are on entirely different scales and it tends to be easier to compare results if they’re normalized relative to some established quantity so that all results are on the same scale. VRMSE is used for the same reason, but we feel very strongly the denominator is more appropriate.
>
> We use nMSE in this explanation for cleaner notation. If we look at the denominator of nMSE, we see a clear problem with the idea that nMSE is putting all losses on the same scale. The second moment in the denominator $\mathbb E[y^2]$ admits a variance decomposition into $\mathbb E[y^2] = Var(y) + \mathbb E[y]^2$ (derived from the definition of variance). The mean term, $\mathbb E[y]^2$, is problematic for constructing a loss measure. If we had a nearly constant pressure field distributed $P\sim \mathcal N(10^{10}, 1)$ and a predicted $\hat P \sim \mathcal N(10 + 10^{10}, 1)$, the mean term of the denominator in nMSE would drive the loss to order $10^{-10}$, suggesting the prediction is excellent, even though it is extremely improbable under the true distribution. Variance normalization, on the other hand, scales performance relative to the true spread of the field rather than relative to the magnitude of the mean value (which can be learned entirely through the bias term of the output head) which we feel is more intuitive.
>
> > many cases the surrogate models fail to outperform a mean prediction after the budgeted 12 hours.
>
> One small, but important clarification here is that this is outperforming the mean of the target field rather than the mean of the full dataset - for a slow moving field outperforming this should be trivial while for a rapidly changing field, it could still be difficult. This is something that we will clarify in the text with the metric explanation discussed above.
>
> > I believe there is a typo in the abstract: should "new challenges poses" read "new challenges posed"?
>
> Thanks! Fixed.
>
> ---
>
> Once again, we'd like to thank the reviewer for their time and expertise. We hope that we were able to address your concerns. If any lingering concerns or questions remain, please let us know!

---

### Decision · Program_Chairs · 2024-09-26

**Decision:**

Accept (Poster)

**Comment:**

This paper introduces a dataset of numerical simulations of spatiotemporal physical systems. It includes 16 types of simulations, comprising 15TB of data. The data is split into random training, validation, and test splits, and a variety of baseline models are evaluated.

Reviewers appreciate the wide range of physical phenomena, clarity of writing/visualizations, availability of a PyTorch interface. Some questions were raised regarding the relevance of certain datasets (e.g., 2D vs. 3D turbulent flows) or the applicability of VRMSE as a metric for all datasets. However, most reviewers agree that the strengths of this paper outweigh the limitations and I recommend accepting this paper.